# Epstein-Barr virus reactivation induces divergent abortive, reprogrammed, and host shutoff states by lytic progression

Elliott D. SoRelle[1,2☯¤]*, Lauren E. Haynes[1,2☯], Katherine A. Willard[1,2], Beth Chang[3], James Ch'ng[4], Heather Christofk[4,5], Micah A. Luftig[1,2]*

1 Department of Molecular Genetics and Microbiology, Duke University School of Medicine, Durham, North Carolina, United States of America, 2 Duke Center for Virology, Durham, North Carolina, United States of America, 3 Department of Integrative Immunobiology, Duke University School of Medicine, Durham, North Carolina, United States of America, 4 Department of Biological Chemistry, David Geffen School of Medicine, University of California, Los Angeles (UCLA), Los Angeles, California, United States of America, 5 Jonsson Comprehensive Cancer Center, UCLA, Los Angeles, California, United States of America

☯ These authors contributed equally to this work.
¤ Current address: Department of Microbiology and Immunology, University of Michigan Medical School, Ann Arbor, MI, United States of America
* sorelle@med.umich.edu (EDS); micah.luftig@duke.edu (MAL)

**Data Availability Statement:** We have deposited our single-cell RNA sequencing data with NIH GEO

## Abstract

Viral infection leads to heterogeneous cellular outcomes ranging from refractory to abortive and fully productive states. Single cell transcriptomics enables a high resolution view of these distinct post-infection states. Here, we have interrogated the host-pathogen dynamics following reactivation of Epstein-Barr virus (EBV). While benign in most people, EBV is responsible for infectious mononucleosis, up to 2% of human cancers, and is a trigger for the development of multiple sclerosis. Following latency establishment in B cells, EBV reactivates and is shed in saliva to enable infection of new hosts. Beyond its importance for transmission, the lytic cycle is also implicated in EBV-associated oncogenesis. Conversely, induction of lytic reactivation in latent EBV-positive tumors presents a novel therapeutic opportunity. Therefore, defining the dynamics and heterogeneity of EBV lytic reactivation is a high priority to better understand pathogenesis and therapeutic potential. In this study, we applied single-cell techniques to analyze diverse fate trajectories during lytic reactivation in three B cell models. Consistent with prior work, we find that cell cycle and MYC expression correlate with cells refractory to lytic reactivation. We further found that lytic induction yields a continuum from abortive to complete reactivation. Abortive lytic cells upregulate NFκB and IRF3 pathway target genes, while cells that proceed through the full lytic cycle exhibit unexpected expression of genes associated with cellular reprogramming. Distinct subpopulations of lytic cells further displayed variable profiles for transcripts known to escape virus-mediated host shutoff. These data reveal previously unknown and promiscuous outcomes of lytic reactivation with broad implications for viral replication and EBV-associated oncogenesis.

under GSE272763. All other relevant data are in the manuscript and its supporting information files.

**Funding:** o E.D.S. wishes to acknowledge support from the Department of Molecular Genetics and Microbiology Viral Oncology Training Grant (NIH T32 #T32CA009111) and a postdoctoral fellowship from the American Cancer Society – Charlotte County, Virginia TPAC (PF-21-084-01-DMC). L.E.H. wishes to acknowledge NIH F31 support from the National Institute of Dental and Craniofacial Research (NIDCR; award #1F31DE033216). K.A.W. wishes to acknowledge funding from the National Science Foundation Graduate Research Fellowship Program (NSF GRFP #1644868). J.C. wishes to acknowledge support from St. Baldrick's Foundation. This work was supported by NIH R01 funding from the National Institute of Dental and Craniofacial Research (NIDCR; award #R01DE025994; M.A.L.) and the National Cancer Institute (NCI; award #R01CA215185; H.C.). The funders had no role in study design, data collection and analysis, decision to publish, or preparation of the manuscript.

**Competing interests:** The authors have declared that no competing interests exist.

## Author summary

Viral infections profoundly alter host cell biological programming in ways that potentiate disease. Epstein-Barr virus (EBV) is a particularly prevalent human pathogen associated with diverse cancers and several autoimmune disorders. EBV predominantly establishes latent infection in B cells and can promote B cell malignancies through functions of well-characterized latent oncoproteins. Aspects of the viral lytic cycle also clearly contribute to EBV-associated diseases, although pathologic roles of lytic reactivation are incompletely understood. Here, we use single-cell techniques to examine cellular responses to EBV lytic reactivation in multiple B cell models. Consistent with prior studies, reactivation from latency is incomplete (abortive) in some cells and successful in others. Abortive and full lytic trajectories exhibit distinct biological responses that each may promote pathogenesis and reinforce bimodal latent-lytic control. Intriguingly, a portion of cells that proceed through the lytic cycle exhibits unexpected and striking expression of genes associated with cellular reprogramming, pluripotency, and self-renewal. Collectively, this study provides a valuable resource to understand diverse host-virus dynamics and fates during viral reactivation and identifies multiple modes of EBV lytic pathogenesis to investigate in future research.

## Introduction

Viral infections lead to heterogeneous cell fate outcomes including resistance, abortive infection, latency, or full virion amplification often leading to cell death. Cells that resist viral infection often display elevated pre-existing anti-viral responses [1–4]. Likewise, cell responses that enable survival following virus replication can prime for further anti-viral responses [5,6]. Herpesviruses are large double-stranded DNA viruses that provide a unique and complex infection paradigm to model the heterogeneity of viral infection as they reactivate from a latent state in response to diverse stimuli.

Epstein-Barr virus (EBV) was the first oncogenic human virus to be discovered [7]. Since its isolation from endemic Burkitt Lymphoma (BL) cells in 1964, EBV infection has been linked to an expansive set of human cancers and, more recently, autoimmune diseases [8–11]. EBV infection in immunosuppressed individuals can lead to post-transplant lymphoproliferative disease (PTLD) [12] and HIV-related diffuse large B cell lymphomas (DLBCL) [13] as well as up to 40% of Hodgkin Lymphoma (HL) [14]. and rare individuals with chronic active EBV (CAEBV) can develop T and NK cell lymphomas [15,16]. Beyond these hematologic malignancies, EBV infection is associated with epithelial cancers such as nasopharyngeal carcinoma (NPC) [17] and gastric carcinomas [18]. Collectively, EBV causes is or otherwise associated with nearly 2% of all cancers diagnosed annually [8].

This prevalence in malignant disease vastly underrepresents the success of EBV as a human pathogen. Globally, it is estimated that over 95% of adults are infected with EBV [19]. EBV is transmitted via saliva, which enables the virus to traverse oral epithelial tissues and infect B lymphocytes within the tonsils [20]. EBV infects B cells via the surface receptor CD21 (CR2) [21,22] and rapidly induces B cell adaptive immune programs to mimic germinal center (GC)-like dynamics [23–26]. Successful evasion of antiviral defenses, immune tolerance checkpoints, and growth-induced damage [27–29] allows memory B cells latently infected with EBV to exit from this virus-manipulated GC reaction. Viral latency establishment within the memory B cell compartment yields lifelong persistence [30, 31]. Lytic reactivation from this latent state triggers the production of new virions and is essential to the replicative cycle and transmission

between hosts. The lytic gene program is transcriptionally orchestrated by two immediate early (IE) lytic genes: *BZLF1* (encodes for the transcription factor Zta / Z / ZEBRA) and *BRLF1* (encodes for the transcription factor Rta / R) [32–34]. While Zta and Rta both play essential roles in lytic reactivation, Zta is the master lytic transactivator in B cells. *BZLF1* expression is induced upon cell differentiation and stress [35,36], a prototypical example being post-GC B cell differentiation into plasmablasts [37]. Host cell transcriptional regulators of plasma cell generation including XBP1 and BLIMP1 (*PRDM1*) induce EBV lytic reactivation via direct transactivation of the *BZLF1* promoter [38–40]. Zta then transactivates subsequent expression of early and late lytic genes by binding at Z-responsive elements (ZREs) throughout the viral genome [41]. As an AP-1 family homolog [33], Zta also binds loci throughout the host genome [42] and has characteristics of a 'pioneer' transcription factor. Consistent with this, *BZLF1* expression and the early stages of EBV reactivation cause considerable alterations to the host cell epigenome and resulting gene expression [43,44].

Prior work suggests that lytic gene expression is functionally important for tumorigenesis. Notably, viral strains that carry the NFATc1-responsive Z promoter variant Zp-V3 exhibit increased lytic replication and are enriched in EBV-associated cancers relative to strains with prototypical Zp [45]. In SCID and NSG mouse models with reconstituted human immune systems, significantly fewer animals developed EBV[+] lymphomas after infection with *BZLF1* knockout virus versus a wild-type (WT) control strain [46]. Further, infection with a Zta-over-expressing strain that failed to complete reactivation (i.e., abortive lytic) promoted tumor growth in mice similar to WT EBV [47]. Recent experiments in immunocompromised mice confirmed the tumorigenic role of abortive lytic infection by using EBV lacking the *BALF5* gene, which encodes a viral DNA polymerase subunit essential for lytic replication [48]. These studies demonstrated that expression of *BZLF1* (and possibly other early lytic genes) contributes to tumorigenesis *in vivo* regardless of the potential for horizontal infection of bystander cells by new virions. While detailed insights regarding the oncogenic effects of successful or abortive lytic replication are limited, tumor microenvironment inflammatory conditioning by cytokines secreted from reactivating cells has been proposed [49–53].

Another complication in the relation between viral reactivation and oncogenicity stems from observations that a significant proportion of EBV-infected tumor cells are resistant or otherwise refractory to lytic reactivation. In BL-derived P3HR1 and Akata cells, high expression of the oncoprotein c-Myc promotes viral latency maintenance and suppresses lytic reactivation via direct interaction with the origin of lytic replication (*oriLyt*) and inhibition of chromatin looping to activate *BZLF1* expression [54]. Accordingly, *MYC* suppression facilitates *BZLF1* expression and the subsequent induction of viral lytic genes. It is noteworthy that constitutive oncogene expression favors viral genome propagation through proliferation of latently infected host cells whereas lytic replication becomes a more advantageous strategy in its absence. Similarly, BL-derived cells refractory to lytic reactivation have also been found to express high levels of STAT3 [55–57], which functions as an oncogene in B cells and inhibits apoptosis via induction of BCL2 expression. Beyond simply being expressed by refractory cells, STAT3 antagonizes lytic reactivation of EBV[+] cells through the functions of its transcriptional targets [56]. In fact, LCLs derived from patients with autosomal dominant hyper-IgE syndrome (AD-HIES), a disease that leads to non-functional STAT3 activity, went lytic at a higher rate than LCLs derived from healthy donors [58]. Given the therapeutic potential of drug-induced lytic reactivation followed by viral DNA synthesis inhibition to treat EBV-latent cancers, investigators are actively exploring means to make refractory cells more sensitive to lytic induction [59–61]. However, such efforts should be weighed against the known associations between the EBV lytic cycle and oncogenesis, which remain to be fully elucidated.

Many EBV gene products contribute to virus-driven malignancies by mediating functions associated with cancer hallmarks including uncontrolled proliferation, tumor suppressor inhibition, epigenetic reprogramming, genome instability, apoptotic resistance, and immune evasion [62]. EBV$^+$ cells with cancer stem cell (CSC) features have also been reported in NPC and gastric carcinoma [63, 64], suggesting the potential for cellular self-renewal associated with infection. In the CSC model, a small subset of tumor cells retain the capacity for self-renewal and proliferation through activation of signaling pathways (e.g., Wnt, Notch), transactivators of the epithelial-to-mesenchymal (EMT) transition, and critical regulators of pluripotency (e.g., SOX2, OCT4). CSCs may serve as progenitors for other tumor cells, especially in lymphoid malignancies that are derived from cells of origin that intrinsically retain self-renewal properties to support immunologic memory [65–67]. Aberrant expression of self-renewal genes and other CSC biomarkers [68] may originate from significant (epi)genomic reprogramming and result in cellular phenotypic plasticity. Lytic replication of EBV (and DNA viruses from several other families [69]) clearly constitutes a major reprogramming event for the host cell. Nuclear chromatin is globally disrupted by IE gene expression, the formation of viral replication compartments, and the accumulation of viral DNA [43,70]. Moreover, preferential binding of BZLF1 to methylated promoters can reverse epigenetic silencing of both EBV and cellular genes through nucleosome eviction, resulting in heterochromatin-to-euchromatin conversion [44,71–73]. While evidence for stem-like reprogramming and CSC gene expression during the EBV lytic cycle has not been reported to our knowledge, it is noteworthy that reactivation of HSV-1 (another herpesvirus) induces embryonic development programs including Wnt/β-catenin activity that licenses late viral gene expression [74].

These previous studies demonstrate that EBV reactivation from latency is a complex process that culminates in heterogeneous host cell responses germane to the progression of virus-associated cancers. Single-cell sequencing techniques are particularly well suited to dissect the inherent complexity of host-virus interactions and their effects on cell fate [74–77]. In recent studies of early EBV infection [25,26] and established latency [78,79], we have used single-cell sequencing to successfully resolve and study diverse phenotypes arising from complex host-pathogen dynamics. We reasoned that a similar high-resolution experimental and informatic approach would clarify distinct courses of lytic reactivation, provide essential data for future studies of viral pathogenesis, and inform potential therapeutic strategies to address EBV-driven oncogenesis. To this end, we performed time-resolved single-cell RNA sequencing (scRNA-seq), flow cytometry, and RNA Flow-FISH (fluorescence *in situ* hybridization) in P3HR1-ZHT cells to define initial cell state diversity, differential fate trajectories, and previously unknown lytic response phenotypes. We further extended and validated these studies with scRNA-seq in the B958-ZHT LCL and the Akata BL cell line thereby defining common mechanisms across diverse viral strains and reactivation stimuli.

## Results

### Heterogeneous responses to EBV lytic reactivation in individual cells

P3HR1-ZHT cells are an inducible model of EBV lytic reactivation (**Fig 1A**). This model system constitutively expresses the EBV immediate early lytic transactivator Zta (encoded by the *BZLF1* gene) fused with a modified murine estrogen receptor hormone binding domain. While the encoded fusion protein is normally rapidly degraded, addition of 4-hydroxytamoxifen (4HT) stabilizes it and promotes its nuclear translocation, whereupon the Zta domain binds and transactivates Zta-responsive elements (ZREs) in both host and viral genomes. Because Zta has positive regulatory control of its own promoter via ZRE binding [80], 4HT treatment also leads to expression of endogenous *BZLF1*, thus initiating viral lytic reactivation.

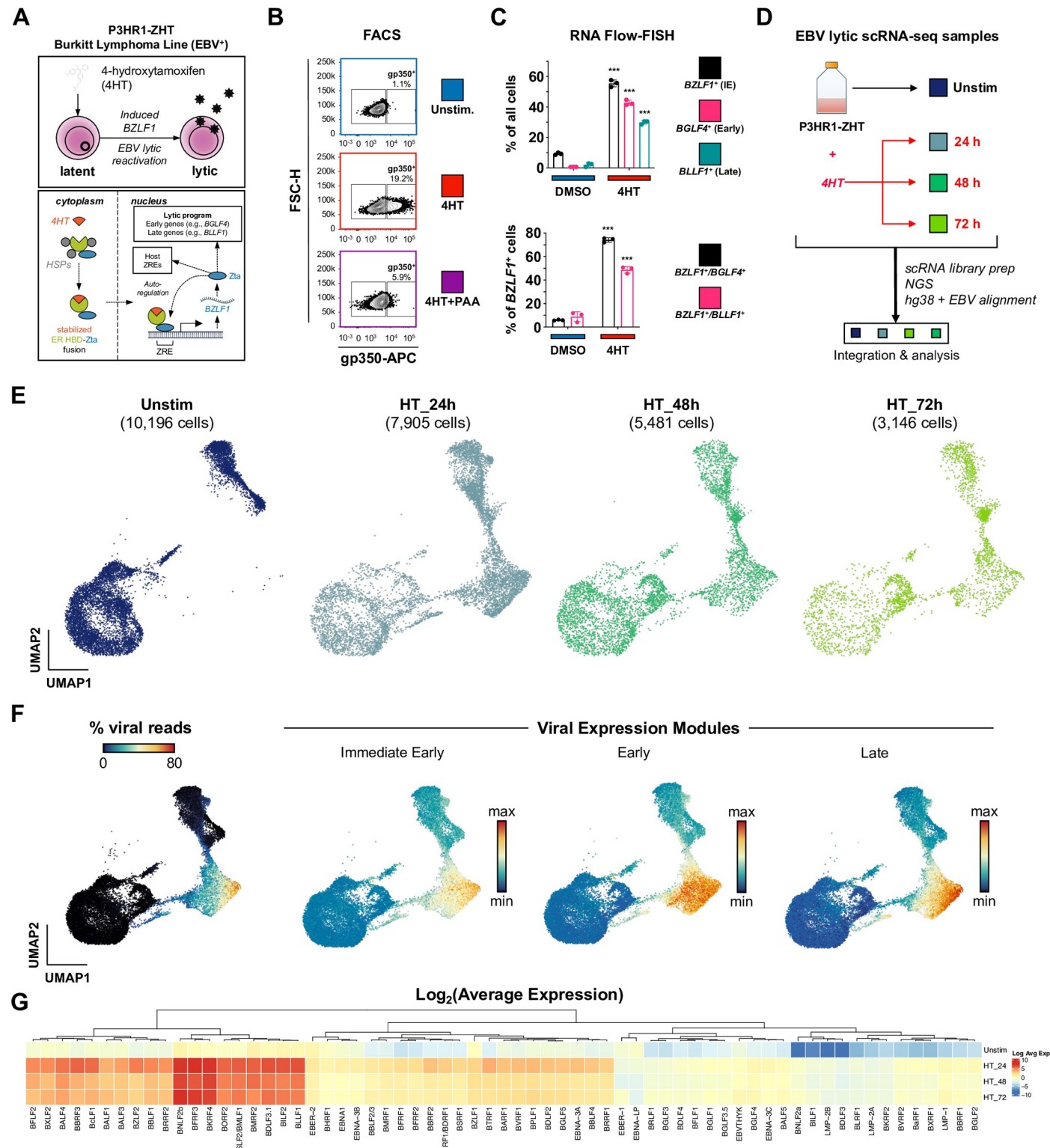

**Fig 1. EBV lytic reactivation in the P3HR1-ZHT Burkitt Lymphoma line at single-cell resolution.** (A) Schematic of 4HT-inducible BZLF1 (Zta) expression initiating lytic reactivation in the Burkitt Lymphoma-derived P3HR1-ZHT cell line. (B) Flow cytometry validation 24 h after 4HT-induced lytic reactivation and inhibition of complete reactivation by phosphonoacetic acid (PAA) in P3HR1-ZHT. Cellular expression of the viral glycoprotein gp350 (encoded by the late lytic gene *BLLF1*) serves as a proxy for successful reactivation. Co-treatment with the viral DNA polymerase inhibitor PAA prevents complete reactivation by blocking viral DNA replication, which is required for expression of late viral genes / gene products including gp350. (C) RNA Flow-FISH validation of select immediate early

(IE), early, and late lytic gene expression in P3HR1-ZHT. The majority of cells express detectable *BZLF1* 24 h after 4HT treatment. Substantial fractions express early genes including the EBV DNA polymerase (*BGLF4*) and late genes including *BLLF1*. However, not all *BZLF1*⁺ cells exhibit early and late gene expression, indicating variable progression of reactivation in individual cells. Asterisks denote significantly higher expression in 4HT-treated samples versus DMSO controls (n = 3 per condition; two-tailed Welch's t-test; \*\*\*p<0.001). (D) Experimental design schematic for time-resolved scRNA-seq study of EBV reactivation in P3HR1-ZHT. Single-cell libraries were prepared from unstimulated cells and from cells at three timepoints (24 h, 48 h, and 72 h) after 4HT treatment. Libraries were sequenced, mapped to a multispecies reference genome, integrated into a single data object, and analyzed. (E) UMAP representation of single cells captured across the experimental timecourse. Plots display the number of cells in each library after QC filtering. (F) EBV gene expression overview in merged timecourse scRNA-seq data. (From left to right) Viral fraction of captured transcripts per cell; scores for an immediate early (IE) expression module (*BZLF1*, *BRLF1*); scores for an early gene expression module (*BRRF1*, *BBLF4*, *BALF1*, *LF3*, *BARF1*, *BaRF1*, *BVLF1*, and *BALF3*); scores for a late gene expression module (*BZLF2*, *BLLF1*, *BILF2*, *BBRF3*, *BcLF1*, *BRRF2*, *BSRF1*, *BCRF1*, and *BBRF1*). Modules were curated based on viral expression kinetics determined by CAGE-seq [162]. (G) Hierarchically clustered average expression of all detected viral genes by timepoint.

Although all cells in the P3HR1-ZHT line express the inducible construct, it has been observed that complete EBV lytic reactivation occurs only in a subset of 4HT treated cells [81,82].

We confirmed inducible yet non-uniform viral reactivation of P3HR1-ZHT cells in response to 4HT treatment using FACS staining for the viral glycoprotein gp350, which was expressed in cells that reached the late stage of lytic reactivation. Unstimulated P3HR1-ZHT cells expressed minimal gp350 (1.1%), but treatment with 100 nM 4HT for 24 hours resulted in gp350 expression in 19.2% of cells. When we simultaneously treated cells with 4HT and PAA, an inhibitor of viral DNA replication, we observed a significant reduction in gp350 expression by 24 hours (**Figs 1B and S1**). These results indicated that cells exhibited heterogenous responses to viral lytic reactivation and that completion of the full lytic cycle was dependent upon successful viral DNA replication, which has been previously described in herpesviruses [83–87]. We expanded upon these gp350 FACS results using RNA Flow-FISH assays to detect viral RNAs from genes expressed at different stages of the lytic cycle: the immediate early lytic gene *BZLF1*, the early lytic gene *BGLF4*, and the late lytic gene *BLLF1*. After 24 hours of 4HT treatment, we observed a significant increase in expression of all three lytic transcripts compared to mock treated cells. However, there was a stepwise decrease in expression level between early and late lytic genes (**Figs 1C and S2**). These results confirmed that a significant proportion of Z-HT induced P3HR1 cells were refractory to full lytic reactivation.

Since we observed heterogeneous responses upon lytic reactivation, we applied time-resolved single-cell RNA sequencing (scRNA-seq) to study the concurrent cellular responses in the P3HR1-ZHT system after 24, 48, and 72 hours of 4HT treatment compared to untreated cells (**Fig 1D**). UMAP projection of samples by timepoint demonstrated that substantial transcriptomic changes occurred after 4HT stimulation (**Fig 1E**). Cells expressing high levels of viral reads clustered together, however there was a distinction between cells expressing immediate early, early, and late viral transcripts (**Fig 1F**). Analysis of all EBV transcripts identified genes with high, moderate, and low expression; however, all 4HT-treated samples expressed more viral transcripts compared to untreated cells (**Fig 1G**). These results confirmed heterogeneous responses to lytic reactivation observed by flow cytometry and enabled subsequent genome-wide analyses.

## Identification of distinct EBV reactivation response clusters

Cells from integrated timecourse scRNA-seq libraries were hierarchically clustered by host and viral transcriptome similarity, which led to the identification of five main clusters (**Fig 2A**). Unstimulated cells were mostly present in clusters A and B, while clusters C, D, and E primarily comprised 4HT-treated cells across the experimental time course (**Fig 2B**) and displayed elevated viral gene expression compared to clusters A and B (**S3 Fig**). Further examination of these clusters revealed differences in the number of total and unique RNAs, the percentage of viral RNAs, and the percentage of mitochondrial RNAs (**Fig 2C**). These

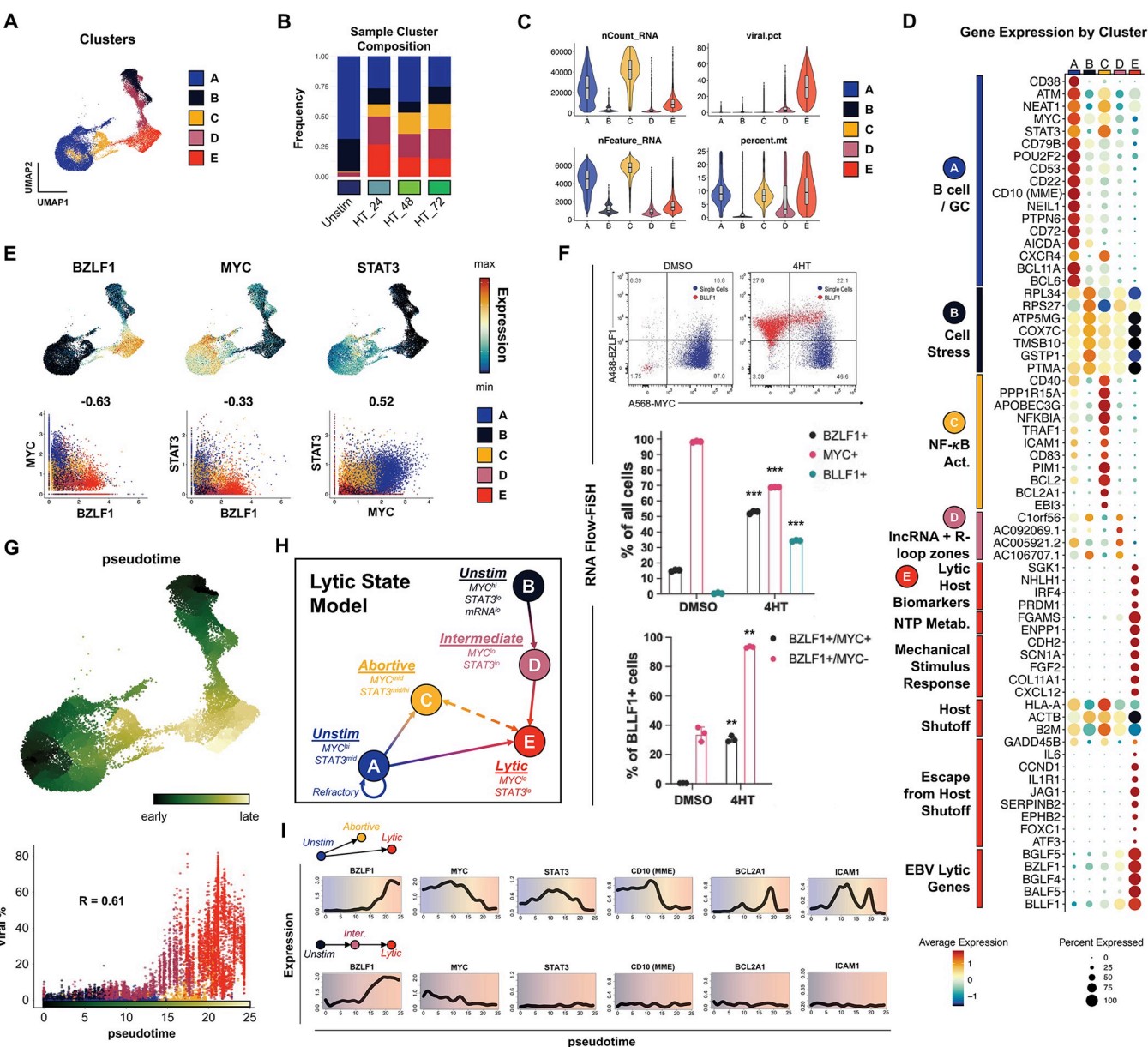

**Fig 2. P3HR1-ZHT phenotypic heterogeneity and response trajectories during lytic induction.** (A) P3HR1-ZHT cell clusters identified in merged timecourse data via unsupervised methods. (B) Cluster composition of cells from individual timepoints. Cluster colors are coded as in 2A. (C) QC feature distributions by cluster. The total number of mapped reads per cell is given by nCount_RNA. The number of unique RNA features (i.e., genes, lncRNAs) per cell is given by nFeature_RNA. The viral fraction of mapped reads per cell (viral.pct) and mitochondrial transcript fractions (percent.mt) were calculated using the *PercentageFeatureSet()* function in Seurat [154]. (D) Differential gene expression by cluster. Genes are annotated by their known biological roles and functions derived from gene ontology (GO) analysis and primary literature. Dot size represents the percentage of cells in each cluster that express a given gene and color encodes average expression across the cluster. (E) UMAP expression profiles (top row) and pairwise correlation plots (bottom row, Pearson R) for *BZLF1*, *MYC*, and *STAT3*. Correlation plots depict individual cells colored by cluster. (F) RNA Flow-FISH validation of reduced *MYC* expression in *BZLF1⁺BLLF1⁺* cells (top panel). Asterisks in the middle panel bar plot denote significantly reduced frequency of *MYC⁺* P3HR1-ZHT cells and increased frequencies of *BZLF1⁺ and BLLF1⁺* cells after 4HT treatment (n = 3 per condition; two-tailed Welch's t-test; ***p<0.001). Asterisks in the bottom panel bar plot denote significantly increased frequencies of *BZLF1⁺MYC⁺* and *BZLF1⁺MYC⁻* cells after 4HT treatment (n = 3 per condition; two-tailed Welch's t-test; **p<0.01). (G) UMAP of graph-based pseudotime trajectory calculation for timecourse-merged scRNA-seq data. Trajectory root cells were selected from both clusters A and B, which were present in the unstimulated (day 0) P3HR1-ZHT library (top panel). Viral read content in individual cells ordered by pseudotime and coded by cluster (bottom panel). (H) Cluster- and pseudotime-informed annotated cell state model of EBV lytic reactivation in P3HR1-ZHT. Solid line arrows denote cell response trajectories supported by time-resolved scRNA-seq data. The dashed line denotes a putative state interconversion. (I) Gene expression dynamics along distinct pseudotime trajectories in the lytic reactivation timecourse. Highlighted genes were selected from those differentially expressed across unstimulated, abortive, and fully lytic cells.

differences in unique and total RNA features suggested major phenotypic differences both in unstimulated and reactivated cells. Therefore, we scored the clusters based on cell cycle state and found that there was a decrease in $G_2/M$ specific gene expression and an increase in G1 gene expression after 24 hours of 4HT treatment, consistent with EBV lytic reactivation occurring in a pseudo-S phase [88, 89] (**S4A Fig**). We confirmed this finding using BrdU/7-AAD staining of untreated versus 4HT-treated cells (**S4B Fig**). Consistent with induced cell cycle arrest, lytic reactivation upon 4HT treatment led to a reduction of S phase cells (43.2% vs. 54.3%) and modest increase in $G_0/G_1$ cells. Because pulsed BrdU staining does not discriminate cellular and viral DNA synthesis, a portion of S phase 4HT-treated cells were likely undergoing viral but not cellular DNA synthesis. This was further evidenced by a significant fraction of gp350$^+$ cells within the gated S phase population (**S4B Fig**). We also assayed MitoTracker signal stratified by gp350 expression and found that gp350$^+$ cells had lower mitochondrial content (**S4C Fig**).

## Cells traverse heterogeneous biological response trajectories during lytic reactivation

Next, we analyzed differentially expressed genes by cluster and grouped them by ontology using a combined approach with software-based annotation tools [90] and primary literature searches (**Fig 2D**). Unstimulated cells were almost exclusively present in clusters A and B, which were distinguished from each other by total transcripts and unique features per cell (**Fig 2B and 2C**). Unstimulated cells with high RNA and feature counts (cluster A) exhibited a germinal center (GC) B cell profile including *MME (CD10)* [91], *BCL6* [92,93], *BCL11A* [94], *POU2F2 (OCT2)* [95], and *AICDA (AID)* [96,97]. Along with high *MYC* expression, this phenotype is consistent with the profile of endemic BL from which P3HR1-ZHT is derived. In contrast, unstimulated cells with low RNA and features counts (cluster B) exhibited a cell stress expression signature that included slight enrichment of genes for ribosomal subunits (*RPL34*, *RPS27*), nuclear-encoded components of mitochondrial respiratory complexes (*COX7C*), and the apoptotic resistance genes *PTMA* [98] and *GSTP1*, the latter of which also mediates oxidative stress [99]. Cluster C, which was comprised of 4HT-treated samples, displayed antiviral restriction (*APOBEC3G*, *PPP1R15A*, *TRIM14*, *FURIN*), inflammatory (*CCL4L2*, *CCL3L1*, *NKG7*), and NF-κB signaling (*NFKBIA*, *ICAM1*, *CD83*, *BCL2*, *BCL2A1*) signatures. Cluster D had a similar gene expression pattern to cluster B with the addition of lytic transcripts and several long noncoding RNAs from R-loop "hot spots" (*C1orf56*, *AC092069.1*, *AC005921.2*, *AC106707.1*) associated with genomic instability related to unscheduled gene expression or DNA synthesis (in contexts including herpesviral reactivation) [100–104]. Finally, cluster E primarily contained cells that had entered the lytic cycle after 4HT treatment. Lytic cells expressed known host biomarkers of reactivation (*SGK1*, *NHLH1*, *PRDM1*) [105], downregulation of genes targeted by virus-induced host shutoff (*HLA-A*, *ACTB*, *B2M*) [106] mediated by EBV BGLF5 [107], expression of genes that escape host shutoff (e.g., *GADD45B*, *IL6*, *CCND1*, *JAG1*, *SERPINB2*, *FOXC1*, *ATF3*) [108–110], and numerous IE, early, and late lytic genes. Furthermore, an extended list of EBV genes expressed across all clusters highlights that most lytic transcripts are highly expressed in cluster E, while there is some limited expression of select genes in clusters C and D (**S6 Fig**).

We next focused on individual genes that are differentially expressed between the clusters. We specifically chose *STAT3* and *MYC* because they have been established as key regulators of EBV lytic reactivation [54,56,57,59] (**Fig 2E**). In line with these published results, *MYC* expression was strongly anti-correlated with *BZLF1* induction (**Fig 2E**, bottom left panel). *STAT3* expression, which has been previously shown to be upregulated in cells refractory to lytic

reactivation [59], was likewise anti-correlated with expression of *BZLF1* (**Fig 2E**, bottom middle panel). *STAT3* and *MYC* expression were positively correlated and highest in unstimulated (cluster A) and abortive (cluster C) cells (**Fig 2E**, bottom right panel). Prediction of transcription factor activities based on gene regulatory network (GRN) enrichment likewise identified enhanced STAT3 (and NF-κB) target expression in cluster C (**S5 Fig**). RNA Flow-FISH detection of *BZLF1* and *MYC* validated scRNA-seq data and provided additional insight with respect to partial versus complete reactivation indicated by expression of the late lytic gene *BLLF1* (**Fig 2F**). Specifically, 4HT treatment induced significant increases in *BZLF1*+ and *BLLF1*+ cells and a concomitant decrease in *MYC*+ cells relative to DMSO-treated controls (**Fig 2F**, top and middle panels). Moreover, the majority of *BLLF1*+ cells were *BZLF1*+/*MYC*- (**Fig 2F**, bottom panel).

Given the observed heterogeneity of phenotypic states before and after lytic induction, we aimed to better understand the distinct response trajectories of EBV-infected cells using pseudotemporal ordering (**Fig 2G**). Pseudotime analyses [111] are preferable over purely chronologic sampling for studying biological state transitions due to initial state variability and asynchronous responses to infection among individual cells [26]. Root cells (pseudotime = 0) for the reactivation trajectory graph were chosen within clusters A and B since both of these phenotypes were represented by unstimulated cells (**Fig 2G**, top panel). As shown by per cell viral fractions of captured mRNA transcripts, reactivation generally progresses in pseudotime, with limited viral expression in abortive cells at intermediate coordinates and high viral expression in fully lytic cells in late pseudotime (**Fig 2G**, bottom panel). Notably, trajectories from both clusters A and B pass through incomplete reactivation states (C and D, respectively) before convening within the lytic phenotype (cluster E) at late pseudotime (**Fig 2G**).

Collectively, cluster-resolved expression, *MYC* and *STAT3* profiles, and pseudotime trajectory analysis enabled us to construct a state model for lytic reactivation in the P3HR1-ZHT system (**Fig 2H**). Unstimulated cells express elevated *MYC* and *STAT3* and may undergo abortive reactivation in response to 4HT in which *BZLF1* expression is minimal while *MYC* and *STAT3* levels are largely maintained. Alternatively, cells may proceed to lytic reactivation, during which both *MYC* and *STAT3* expression are severely diminished. Although clusters C and E were connected by a bridge of cells in the UMAP embedding, we cannot make definitive conclusions from these data alone regarding possible interconversion between abortive and lytic states. While global mRNA levels decrease along the transition from A to E consistent with host shutoff, the trajectory from cluster B (unstimulated) through D (intermediate) toward E (lytic) was characterized by relative increases in total and unique host and viral mRNA content. However, reduced *MYC* expression was also observed along the B to E trajectory. Overall, these results indicated that heterogeneity in unstimulated cells and differential responses to *BZLF1* induction each contributed to the generation of distinct cell states during lytic reactivation. Analysis of gene expression along state-specific pseudotime trajectories captured these distinct biological response coordinates (**Fig 2I**). For example, trajectories starting from clusters A and B both exhibited upregulated *BZLF1* and net *MYC* reduction. However, *STAT3* expression was consistently low across B, D, and E while *STAT3* increased from A to C and decreased from A to E. Likewise, dynamic expression of GC B cell and NF-κB signature genes along A➔(C)➔E were not observed along B➔D➔E.

## Abortive lytic cells are characterized by high NF-κB pathway gene expression

Abortive lytic replication, or the initiation of the lytic cycle without expression of late lytic genes / proteins, has been identified in various systems [47,112]. We sought to characterize

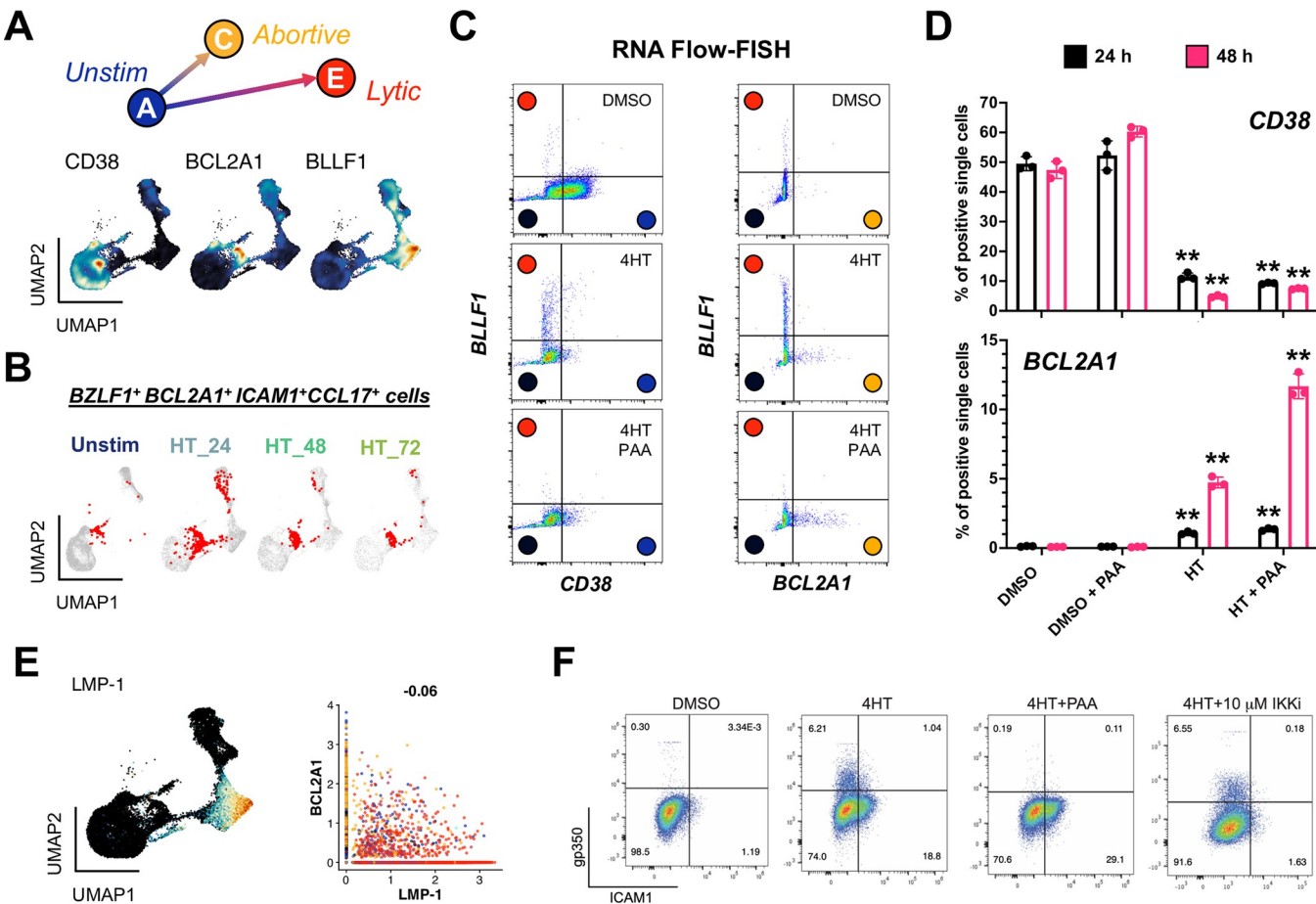

**Fig 3. Validation of an abortive response with elevated NF-κB activity distinct from full lytic reactivation.** (A) Identification of *CD38*, *BCL2A1*, and *BLLF1* as respective biomarkers for unstimulated, abortive, and lytic P3HR1-ZHT cells. (B) Co-detection of *BZLF1* and NF-κB pathway transcriptional targets in abortive cells (co-positive cells in red) by timepoint. (C) RNA Flow-FISH validation of full (*BLLF1*[+]) and abortive (*BCL2A1*[+]) reactivation as orthogonal responses at 48 h post 4HT treatment. DMSO control-treated cells are predominantly *CD38*[+] and exhibit minimal spontaneously lytic (full or abortive) cells (top panel). 4HT treatment induces distinct full lytic and abortive subsets (middle panels). Inhibition of viral DNA synthesis with PAA blocks full lytic reactivation and increases the frequency of *BCL2A1*[+] abortive cells (bottom panels). Colored circles denote predicted corresponding model states defined from scRNA-seq. (D) Frequencies of *CD38*[+] and *BCL2A1*[+] cells presented in 3C by treatment condition at 24 h and 48 h. Asterisks denote significantly decreased frequencies of *CD38*[+] cells and increased frequencies of *BCL2A1*[+] cells upon 4HT and 4HT+PAA treatment versus respective control treatments (n = 3 per condition; two-tailed Welch's t-test; **$p<0.01$). (E) EBV *LMP-1*, which encodes a potent activator of NF-κB signaling, is expressed in late lytic cells (left panel) but not associated with abortive cells that exhibit upregulated NF-κB transcriptomic signature including *BCL2A1* (right panel, Pearson R = -0.06). (F) Flow cytometry analysis of protein biomarkers of full lytic reactivation (gp350) and NF-κB activity (ICAM1) at 48 h post 4HT treatment. Consistent with mRNA measurements, separate gp350[+] and ICAM1[+] populations are induced following 4HT treatment. Co-treatment with PAA reduces gp350[+] cell frequency and increases ICAM1[+] fractions. IKK inhibitor co-treatment reduces ICAM1[+] cell frequency but does not substantially affect gp350[+] cell frequency.

this replication sub-state further through analysis of the abortive lytic cells in the cluster C phenotype. Using markers identified in **Fig 2D** we were able to clearly distinguish unstimulated, abortive lytic, and lytic cells using *CD38*, *BCL2A1*, and *BLLF1* expression, respectively (**Fig 3A**). *STAT3*[+] cells in the *BZLF1*[+] abortive lytic state (cluster C) notably co-expressed *BCL2A1* and other NF-κB pathway target genes (**Fig 3B**). RNA Flow-FISH for *CD38*, *BCL2A1*, and *BLLF1* in cells treated with DMSO (control), 4HT (lytic), and 4HT + PAA (an abortive lytic model due to inhibited viral DNA synthesis) confirmed these distinct response states (**Figs 3C and S7**). This experiment confirmed that *CD38* RNA was primarily expressed in unstimulated cells and decreased upon 4HT treatment. *BLLF1* (gp350) RNA was almost exclusively expressed in 4HT treated cells, and its expression was blocked upon PAA treatment as

expected. *BCL2A1* RNA was significantly elevated in 4HT + PAA-treated cells, especially by 48 hours post-treatment (**Fig 3D**). Thus, these markers reliably delineated latent, abortive, and lytic phenotypes identified from scRNA-seq as clusters A, C, and E.

Because EBV LMP-1 partially mimics the activated CD40 receptor that induces NF-κB signaling, we reasoned that LMP-1 might be associated with the abortive lytic phenotype. However, *LMP-1* expression was largely restricted to cluster E (**Fig 3E**), consistent with its transcription during the lytic cycle [113,114]. This observation suggested that the abortive lytic phenotype and associated NF-κB signaling was not dependent upon *LMP-1* expression. We confirmed this finding through FACS detection of gp350 (lytic cells) and ICAM1, a surface-expressed proxy for NF-κB pathway transcriptional activation (and in this context, abortive reactivation). Untreated P3HR1-ZHT cells did not express gp350 or ICAM1 (**Figs 3F and S8**). Treatment with 4HT induced expression of both gp350 and ICAM1; notably, expression of these proteins was observed in distinct cell subpopulations, supporting our finding that NF-κB signaling was primarily active in cells that had not entered the full lytic cycle. Accordingly, co-treatment with 4HT + PAA to induce an abortive lytic state by blocking viral DNA synthesis led to increased ICAM1$^+$ cell frequency consistent with the *BCL2A1* upregulation observed in **Fig 3D**. Conversely, co-treatment with 4HT and an inhibitor of IKKβ (a key component of NF-κB signaling) eliminated ICAM1 expression, but did not increase gp350 expression. These results demonstrated that NF-κB signaling is a feature of abortive lytic cells that is independent of LMP-1 activity, but does not restrict late viral gene expression.

## Lytic subpopulations are reprogrammed to stem-like plasticity during EBV reactivation

We next focused on the lytic fate by analyzing cells in cluster E. Paradoxically, lytic cells in cluster E collectively expressed the most unique genes (i.e., transcript diversity) of any cluster despite having low mRNA density per cell consistent with host shutoff (**Fig 4A**). In addition to differences in early and late lytic gene expression across cluster E (**Fig 1F**), this observation was consistent with enhanced cell-to-cell variability in gene expression. We therefore subclustered cells at higher resolution to examine heterogeneity among lytic subpopulations (**Fig 4B**). This yielded three subclusters of *BZLF1*$^+$ cells–one with high late gene expression corresponding to complete reactivation (cluster E1) and two with comparatively lower late gene expression (clusters E2 and E3). Differential expression analysis by subcluster revealed remarkably broad cellular plasticity and developmental pluripotency signatures in E2 and E3 (**Fig 4C**). Although *MYC* was downregulated, the master pluripotency regulators *POU5F1* (*OCT4*), *SOX2*, *KLF4*, *NANOG*, and *LIN28A* were expressed in E2 and E3 [115–120]. Intriguingly, many essential transcriptional regulators of pluripotency exit and germ layer specification were also co-expressed with *BZLF1*$^+$ in E2 and E3 lytic subpopulations. Expression of *ALDH1A1*, *ALPL*, *ITGA6*, *CD44*, *PROM1* (*CD133*), *LGR5*, and *YAP1* upregulated in the E2 and E3 phenotypes was consistent with cancer hallmarks including cell plasticity, self-renewal, and drug-tolerant persistence [68,121–126]. Related to *YAP1* expression, we identified distinct Hedgehog [127], Notch [128], and Wnt [129, 130] signaling pathway signatures in E2 and E3 lytic phenotypes as well as Hippo-independent YAP pathway [131] components reported in cancer. E3 cells also expressed genes encoding several PIWI-like family proteins, which protect germline cell genomes from transposable element insertion, maintain stemness, and are upregulated in some cancers [132–135].

In total, 6,900 of 26,728 cells (25.8%) across all sampled timepoints expressed *BZLF1* transcripts. Co-expression of genes including *ALDH1A1* and *SOX2* in a subset of *BZLF1*$^+$ cells demonstrated an association between cellular plasticity and EBV lytic reactivation (**Fig 4D**).

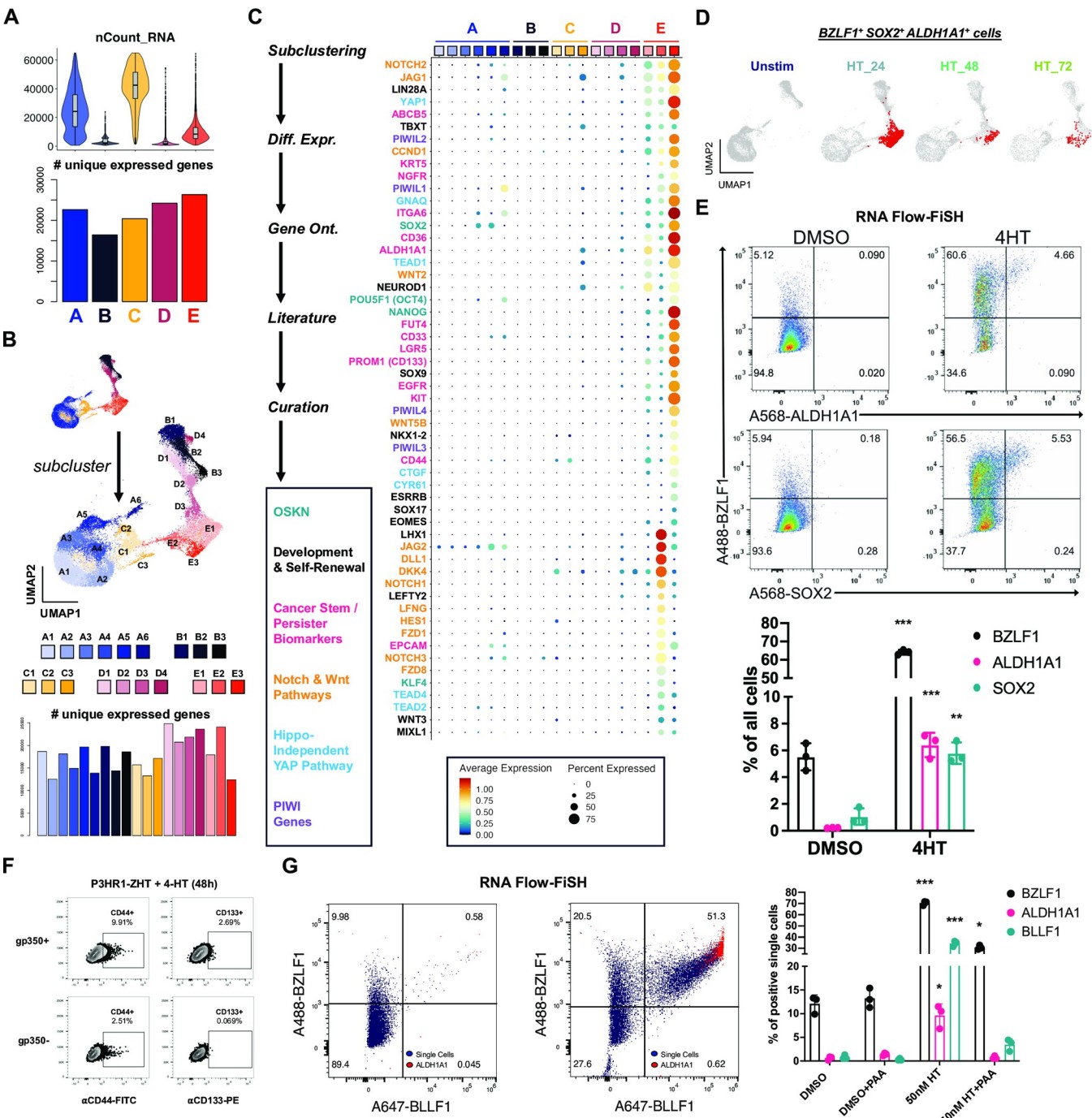

**Fig 4. Cancer-associated cellular plasticity and self-renewal signature identification in EBV lytic cell subsets.** (A) Total mapped RNA reads per cell (top panel) versus total unique genes expressed across each cluster (bottom panel). (B) Unsupervised identification of high-resolution subclusters across P3HR1-ZHT time course scRNA-seq data. (C) Differentially expressed genes upregulated in lytic subclusters (E1, E2, and E3). Genes were identified by comparing each subclusters versus all others, summarized by gene ontology methods, cross-referenced against primary literature, and curated by biological annotation. (D) Co-expression of *BZLF1* and genes associated with cellular pluripotency and cancer stemness (*SOX2, ALDH1A1*) in single cells (co-positive cells in red) by timepoint. (E) RNA Flow-FISH validation of *ALDH1A1* and *SOX2* expression in *BZLF1*+ cells (top panel) at 24 h post 4HT treatment. Frequencies of *ALDH1A1*+ and *SOX2*+ cells significantly increase in response to 4HT induction of the lytic cycle versus DMSO control treatment (bottom panel; n = 3 per condition; two-tailed Welch's t-test; ***p<0.001; **p<0.01). (F) Flow cytometry protein level validation of elevated CD44 and CD133 expression in gp350+ versus gp350- P3HR1-ZHT cells. (G) RNA Flow-FISH analysis of *ALDH1A1* expression by lytic cycle progression at 24 h post 4HT treatment. Rare spontaneously reactivated *BZLF1*+*BLLF1*+ cells express *ALDH1A1* without lytic induction treatment (left panel). The frequency of *BZLF1*+*BLLF1*+*ALDH1A1*+ cells increases upon 4HT treatment (middle panel). *ALDH1A1*+ P3HR1-ZHT cells are significantly enriched after 4HT treatment

but not in the context of co-treatment with PAA to block viral DNA synthesis (right panel; n = 3 per condition; two-tailed Welch's t-test; ***p<0.001; **p<0.01; *p<0.05).

GRN analysis further supported a role for SOX2 transcriptional activity in a fraction of lytic cells (**S10 Fig**). RNA Flow-FISH validated *ALDH1A1* and *SOX2* expression in *BZLF1*+ cells in 4HT treated P3HR1-ZHT cultures (**Figs 4E** and **S9**). To validate these findings at the protein level, we performed intracellular flow cytometry for the SOX2 protein and BMRF1 and confirmed that a small percentage of lytic cells express SOX2 (**S10 Fig**). We also used flow cytometry to validate increased expression of the CSC biomarkers CD44, CD133 (PROM1), and CD166 (ALCAM) at the protein level in gp350+ cells (late lytic) relative to gp350- subsets across treatment conditions (**Figs 4F** and **S11**).

We next examined whether lytic cycle initiation was sufficient to induce CSC-associated pluripotency expression or if successful viral DNA synthesis was required. To do so, we used RNA Flow-FISH to detect *BZLF1*, *ALDH1A1*, and *BLLF1*. *ALDH1A1* was expressed in *BZLF1*+*BLLF1*+ cells following 4HT treatment, consistent with its expression in late stages of lytic reactivation (**Fig 4G**, left and middle panels). Consistent with a role for viral DNA replication in CSC gene induction, co-treatment with PAA and 4HT diminished *BZLF1*+*BLLF1*+ cell frequency and ablated *ALDH1A1* expression (**Fig 4G**, right panel). Collectively, these data support a unique program of cellular plasticity induced in the late phase of EBV lytic reactivation.

## Host shutoff escapees in lytic subclusters exhibit distinct ontologies

Because lytic subclusters identified at high resolution displayed distinct cellular transcriptomes, we asked whether host shutoff responses differed among lytic cells. RNA for *BGLF5*, an early EBV lytic gene that mediates host shutoff [107], was detected at variable levels across lytic cells and inversely correlated with per cell mRNA feature density as expected (**Fig 5A**). We also observed an increase in expression of *BSLF2/BMLF1* (EBV SM) across lytic cells (**S6 Fig**). EBV SM mediates a change from the preferential export of spliced mRNAs to non-spliced viral mRNAs, which leads to an overall reduction in functional cellular gene product expression [136]. Moreover, transcripts for genes previously found to escape host shutoff [109, 110] were identified in each lytic subcluster (E1, E2, and E3) (**Fig 5B and 5C**). Host shutoff escapee expression could be broadly categorized by two patterns–some escapees (e.g., *C19orf66*, *CDKN1B*) were expressed in unstimulated P3HR1-ZHT cells and retained across abortive and lytic cells, whereas other escapees (e.g., *IL6*, *SERPINB2*, *LHX1*, *JAG1*) were exclusively expressed in lytic cells (**Fig 5C**). Intriguingly, lytic subclusters exhibited different host shutoff escapee profiles. Anecdotally, we also noted that several escapees in clusters E2 and E3 were related to inflammatory responses and overlapped with CSC and developmental pluripotency signatures (**Fig 5D**). We applied gene ontology (GO) analyses to differentially expressed genes among lytic subclusters to further investigate potential biological differences. Cells in E2 displayed significant enrichment of GO terms related to mRNA splicing and post-transcriptional regulation and epigenetic regulation versus cells in E3 (**Fig 5E**, top panel). RNA processing GO terms were also upregulated in E2 when compared jointly against clusters E3 and A to filter out differences related to transcripts basally expressed in unstimulated cells (**Fig 5E**, bottom panel). Conversely, the top enriched GO terms in cluster E3 versus E2 were related to cell-cell adhesion, morphogenesis, and diverse tissue-specific developmental programs (**Fig 5F**). Relatively few cellular GO terms were enriched in fully lytic cells (E1), consistent with extensive host shutoff and predominantly viral gene expression (**Fig 5G**).

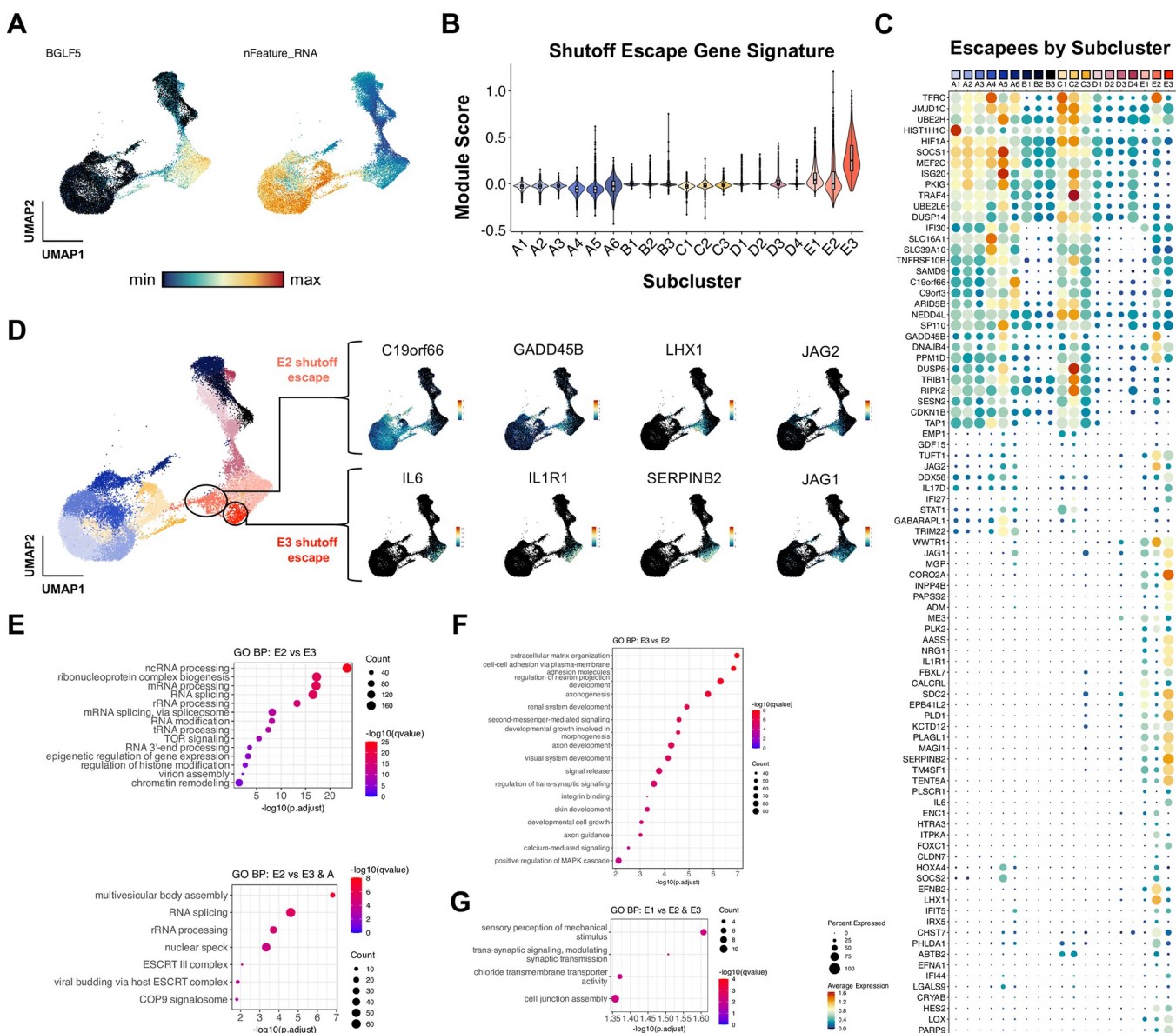

**Fig 5. Distinct virus-mediated host shutoff responses and escapees in lytic subclusters.** (A) UMAP representation of host shutoff mediator *BGLF5* expression (left panel) and per cell feature RNA (right panel) in P3HR1-ZHT timecourse scRNA-seq data. (B) Module scores for a curated set of genes that escape host shutoff (*GADD45B, IL6, CCND1, IL1R1, JAG1, SERPINB2, EPHB2, FOXC1, ATF3, ZNF526, P2RY11,* and *HES4*) by high resolution cluster. (C) Subcluster-level expression of host shutoff escapee genes curated from primary literature. (D) Detail of distinct host shutoff escapee signatures in two lytic subclusters (E2 and E3). (E) Biological process gene ontology (GO) analysis for genes upregulated in lytic subcluster E2 versus E3 (top panel) and E2 versus E3 + A (unstimulated cells). (F) Biological process GO analysis for genes upregulated in lytic subcluster E3 versus E2. (G) Biological process GO analysis for genes upregulated in lytic subcluster E1 versus E2 and E3.

## Phenotype validation across viral strain and host background

Finally, we confirmed key findings through additional independent scRNA-seq experiments capturing responses of B958-ZHT cell lines to 4HT treatment (**Fig 6**), the BL line Akata to anti-Ig stimulation (**Fig 7**), and technical replication in P3HR1-ZHT (**S14 Fig**). Unstimulated and 24 h post-4HT B958-ZHT cell libraries were generated and analyzed as in previous experiments (**Fig 6A**). High-resolution cluster annotations from P3HR1-ZHT scRNA-libraries were

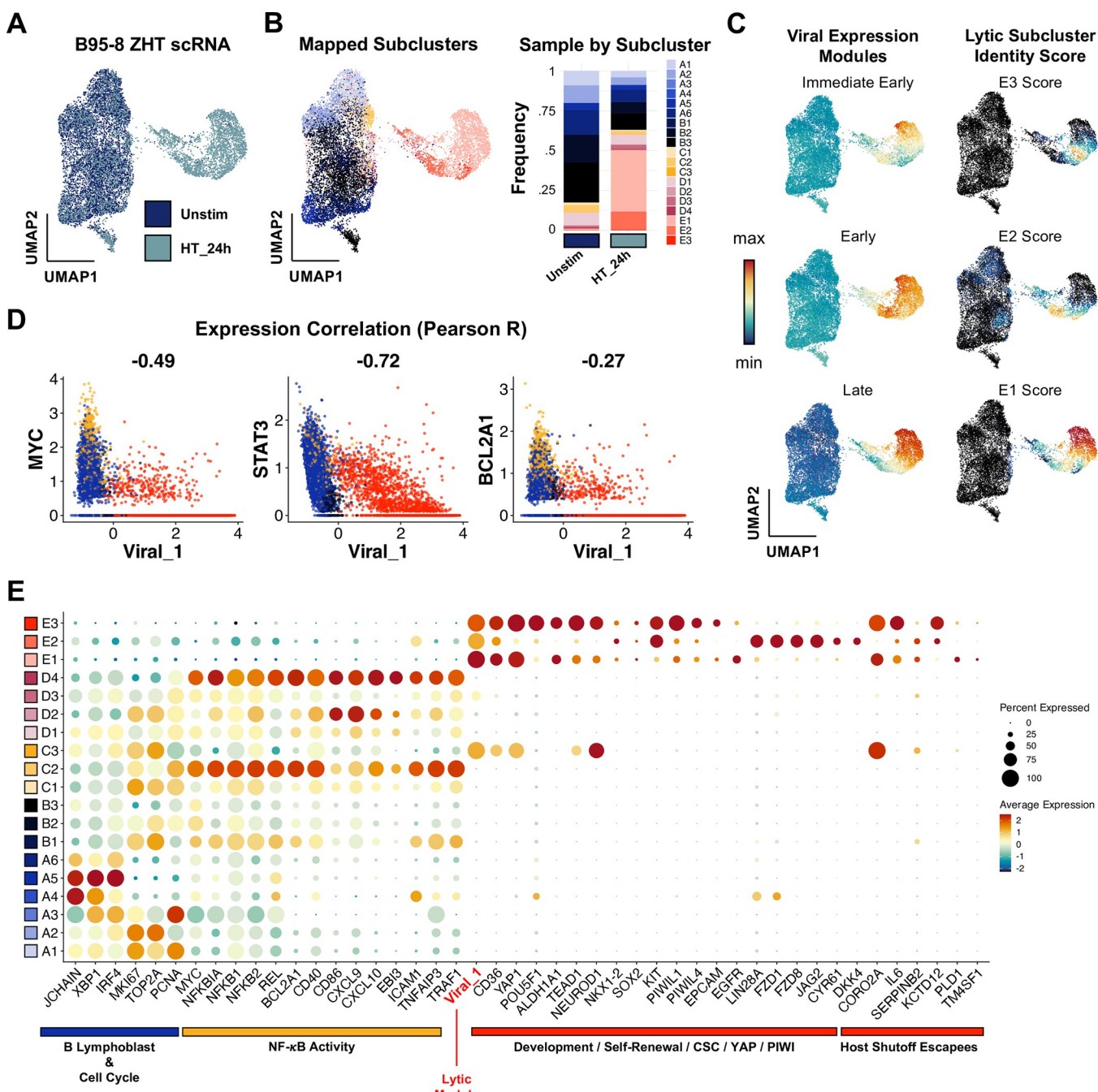

**Fig 6. Lytic subset reprogramming and host shutoff escape signatures are conserved in B958-ZHT lymphoblastoid cells.** (A) UMAP representation of scRNA-seq data from the inducible lytic marmoset lymphoblastoid cell line B958-ZHT before (Unstim) and 24 h after 4HT treatment. (B) Mapping of cell subclusters defined from P3HR1-ZHT analyses to B958-ZHT scRNA-seq data via transfer anchor integration (left panel). Subcluster composition is presented for unstimulated and 4HT-treated cell libraries (right panel). (C) Viral expression module (IE, early, late) and mapped lytic subcluster scores in timecourse merged B958-ZHT data. Of note, the assigned subclusters in 6B represent qualitative classifications based on maximum annotation signature scores for each cell. Accordingly, a given cell may score highly for more than one related signature while being assigned to a single classification. The underlying quantitative signature scores for E1, E2, and E3 presented here thus reflect a lytic phenotypic continuum rather than purely discrete states. (D) Conserved anticorrelation between EBV gene expression (Viral_1 module score) and genes characteristic of unstimulated and abortive phenotypes (*MYC*, *STAT3*, *BCL2A1*). Values denote pairwise Pearson R coefficients. (E) Conservation of key gene expression signatures identified from P3HR1-ZHT (a BL cell line) within B958-ZHT (a lymphoblastoid cell line) during EBV lytic reactivation.

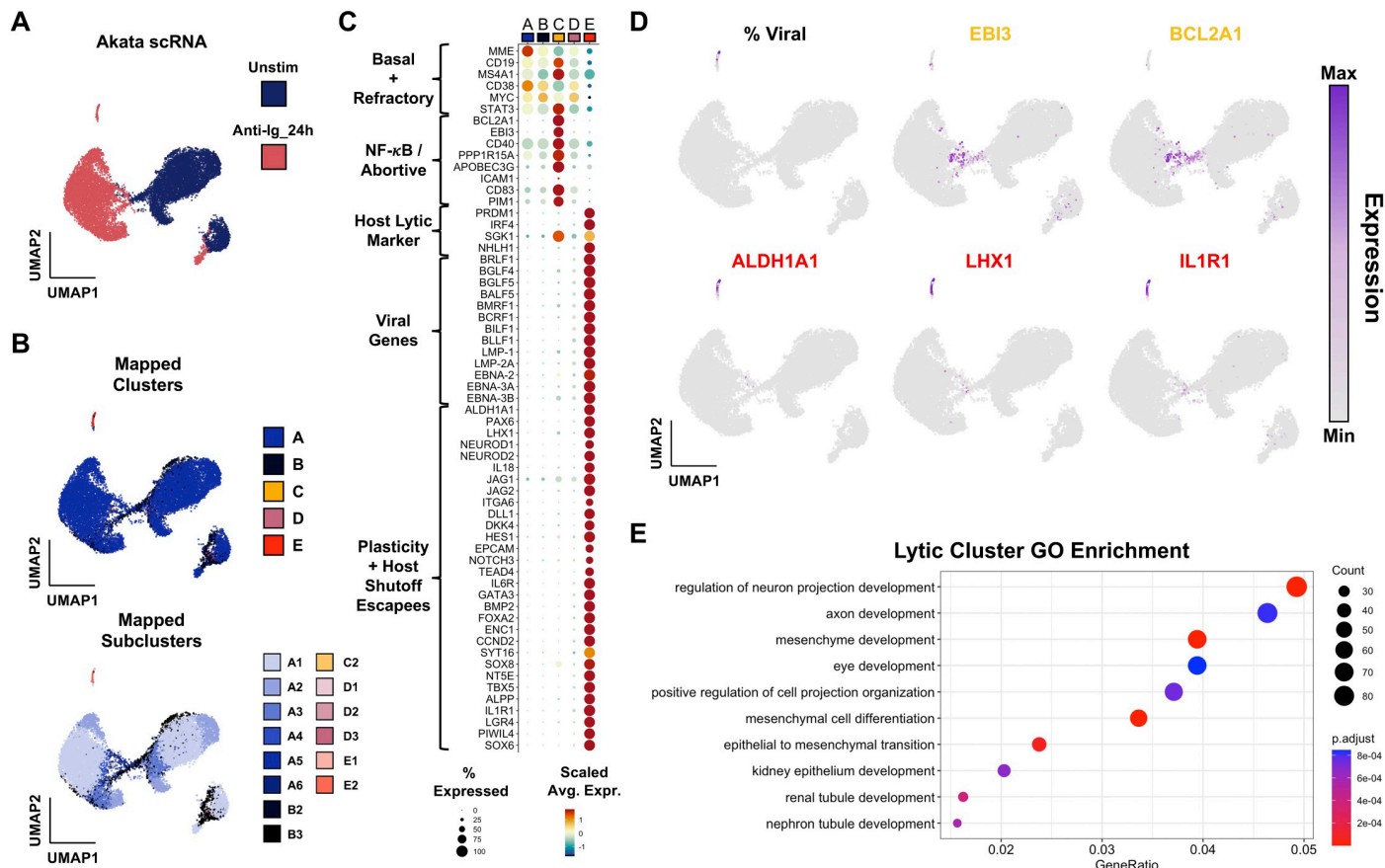

**Fig 7. Validation of abortive and reprogramming gene expression signatures in Akata BL cells stimulated by anti-Ig.** (A) Integrated scRNA-seq data from Akata cells without stimulation (blue) and 24 h after anti-Ig treatment to induce lytic reactivation (rose). (B) Cluster and subcluster mapping from P3HR1-ZHT data to Akata data via transfer anchor integration. Clusters are colored as for P3HR1-ZHT and B958-ZHT datasets. Colors represent the reference phenotype with the maximal integration score for each cell. (C) Cluster-resolved expression in integrated Akata scRNA-seq dataset. (D) UMAP visualization of viral read fraction and representative markers of abortive (yellow) and reprogramming (red) signatures. *LHX1* and *IL1R1* are known escapees of host shutoff. (E) GO enrichment of DE genes upregulated in anti-Ig-induced Akata lytic cells.

mapped to B958-ZHT cells by anchor feature identification and transfer to evaluate the preservation of biological phenotypes across cell systems (**Fig 6B**). Cells corresponding to each high-resolution cluster were identified in the B958-ZHT dataset. Viral IE, early, and late gene expression modules were also scored across B958-ZHT cells and compared against scores for the three lytic subclusters (**Fig 6C**). As in the P3HR1-ZHT system, E1 cells exhibited high late gene expression consistent with complete reactivation while E2 and E3 cells displayed reduced late gene scores. In B958-ZHT, the E3 cluster most closely associated with plasticity and self-renewal signatures had the lowest IE, early, and late expression relative to other cells in lytic clusters. Prior findings of viral gene anticorrelation with *MYC*, *STAT3*, and *BCL2A1* (**Fig 6D**) and lytic cell upregulation of cancer-associated stem-like pluripotency and host shutoff escapees were conserved in B958-ZHT (**Figs 6E, S12, and S13**).

In Akata cells, unstimulated and 24 h post anti-Ig cell libraries were generated as in previous experiments (**Fig 7A**). We again mapped annotations from the P3HR1-ZHT scRNA-libraries to Akata cells by anchor feature identification, both at the cluster and subcluster level (**Fig 7B**). As in the P3HR1-ZHT and B958-ZHT systems, *STAT3*, *MYC*, and *BCL2A1* expression were anticorrelated with successful lytic reactivation (**Fig 7C and 7D**). Viral gene

expression and expression of plasticity markers like *ALDH1A1* were co-expressed (**Fig 7C and 7D**). Gene ontology enrichment for differentially expressed genes in the lytic cluster also highlights the presence of plasticity and developmental factors (**Fig 7E**). Thus, our findings in the P3HR1-ZHT system are applicable across EBV strains, host cell genetic backgrounds, and reactivation stimuli.

## Discussion

The single-cell data presented herein substantially expand and refine transcriptome-wide contours of host-virus dynamics during the EBV lytic cycle. Prior studies discovered that EBV-infected BL cells are prone versus resistant to reactivation dependent on STAT3 expression, activity, and functions of its downstream transcriptional targets [56,57,59]. A population of *STAT3$^{-/lo}$* cells in unstimulated P3HR1-ZHT revealed by scRNA-seq (cluster B), which exhibits globally reduced mRNA levels consistent with cellular quiescence, may be more permissive to successful reactivation than cells with basally elevated *STAT3* (cluster A). Additionally, cells that undergo abortive lytic reactivation retain *STAT3* expression (and predicted transcriptional activity) after stimulation, while *STAT3* and host transcript loads are drastically reduced in fully lytic cells, consistent with host shutoff functions exhibited by diverse viruses [137–141]. Single-cell data are also consistent with the functional importance of c-MYC in regulating EBV latency versus lytic reactivation [54]. *MYC* expression exhibits cluster-level patterns similar to *STAT3*, with the notable exception that *MYC* is more strongly expressed in cluster B cells–likely due to constitutive expression resulting from the chr8:chr14 (*Ig-MYC*) translocation in BL. Single-cell sequencing and RNA-FiSH results further identify unique upregulation of NF-κB and IRF3 pathway transcriptional targets in abortive lytic cells. Inhibition of viral DNA synthesis by PAA treatment accentuates this abortive phenotype, though we recognize that this model may not truly recapitulate an abortive lytic state *in vivo*. Regardless, paired with STAT3 and MYC activity, we speculate that this concerted response might sustain viability and reinforce latency in cells that fail to meet the lytic switch threshold.

Acquisition of cellular plasticity within lytic cell subsets in multiple EBV$^+$ B cell models is particularly striking. Several aspects of the lytic cycle could conceivably contribute to host cell plasticity through reversing epigenetic repression of lineage-ectopic genes. As observed across several DNA virus families, EBV genome replication within intranuclear compartments induces dramatic reorganization of host chromatin [69,70,73,142]. Along with this alteration to nuclear architecture, Zta binding at accessible AP-1 recognition sequences [33] (particularly methylated sites [71,72]) may reverse epigenetic silencing through supporting nucleosome eviction, enhancement of chromatin accessibility, and recruitment of transactivators to facilitate aberrant gene expression [43, 44]. ChIP-seq for Zta has revealed many such potential sites throughout the host genome, including *POU5F1 (*Oct-4*)* [42,43]. From the viral perspective, Zta binding across the cellular genome may function as a "sink" that supports bimodal control of the switch between latency (Zta absence or noise-level expression) and lytic reactivation (high Zta) [43]. From the host perspective, our findings suggest that these BZLF1 interactions with cellular DNA and nuclear chromatin remodeling during later stages the lytic cycle have substantial–and potentially pathogenic–collateral effects on biological reprogramming. Along these lines, developmental reprogramming associated with Wnt/β-catenin signaling has been observed in a single-cell study of HSV-1 lytic infection [74].

Additionally, DNA damage, antiviral nucleic acid sensing, cytoskeletal rearrangements, and other major mechanobiological changes that manifest during reactivation may activate intrinsic responses to cellular injury leading to NF-κB and IRF3 signaling in the abortive population [143–145]. Paired with lytic-mediated growth arrest [146,147], we speculate that this process

may engage cellular senescence and injury responses that promote autocrine and paracrine cellular reprogramming. An essential feature of damage-associated induction of cellular pluripotency is upregulation of pro-inflammatory cytokines such as IL-6 [148]. In both P3HR1-ZHT and B958-ZHT scRNA-seq datasets, *IL6* expression was exclusive to fully lytic cell subsets. However, *IL6R* was expressed in abortive cells in P3HR1-ZHT and most latently infected cells in B958-ZHT. In the Akata scRNA-seq dataset, *IL6R* was exclusively expressed in the lytic cluster. Expression *of JAK1/*2 and *STAT3* in latently infected cells from both lines was suggestive of an IL-6 response axis (IL6(R)/JAK/STAT3) known to be activated in hematologic malignancies [149]. This raises the intriguing possibility that cells from one reactivation trajectory and viral replication mode (fully lytic cells) might reinforce the survival and proliferation of tumor cells resulting from an alternative response (abortive, latently infected) through paracrine mechanisms. In addition to its escape from host shutoff [110], IL-6 autocrine support for latent EBV$^+$ B cell proliferation and its depletion in BZLF1- and BRLF1-deificient tumors in murine models of EBV-driven lymphoproliferative disease are especially noteworthy [53,150,151]. A similar effect has been observed during infection with KSHV, which encodes a viral IL-6 homolog. Thus, the developmental pluripotency profiles and responses of lytic cell subsets may be associated with cellular DNA damage responses that have inadvertent pathogenic effects in EBV$^+$ tumors. Notably, cytokine production by EBV-infected tumor cells (including abortive lytic cells) has also been proposed to support oncogenesis through microenvironment conditioning, polarization of tumor infiltrating lymphocytes, and evasion of T-cell surveillance [49,50].

In summary, our findings support a model of differential response trajectories to EBV lytic induction. The first determinant in this model is initial cell state, where ground-state *STAT3* and *MYC* expression and activity predict a 'high-resistance', low-probability path to full reactivation. Conversely, cells with globally reduced transcription and reduced expression of *STAT3* (and *MYC*) at the time of lytic reactivation traverse a 'low-resistance' path with high probability of complete reactivation. These data have potentially important clinical implications, as they suggest that quiescent EBV$^+$ tumor cells may be more sensitive to lytic induction therapies. However, a critical second fate determinant that manifests in lytic cells may complicate this pursuit. To this point, our scRNA-seq and RNA Flow-FISH results are consistent with the previously identified role of lytic cycle induction in tumorigenesis [46,47,53]. Most cells that undergo full reactivation and new virion release are likely to die. However, some lytic cells undergo profound reprogramming to plastic CSC-like states that may promote malignancy through multiple mechanisms, even independent of their own survival. For example, we found transcript-level evidence that lytic cells could reinforce viral latency and survival of abortive or refractory cells via IL6/JAK2/STAT3 signaling. Additional studies are necessary to explore, dissect, and therapeutically perturb the IL-6/JAK/STAT3 pathway in EBV$^+$ lymphomas.

Furthermore, it is important to note that while there were many overlapping findings between our three datasets, there are key differences to be acknowledged for studies moving forward. The P3HR1-ZHT model is a BL-derived line with constitutive MYC expression that lacks functional EBNA2. The B958-ZHT line was transformed *in vitro* with a strain of EBV that contains functional EBNA2. These differences will lead to natural variation in phenotypes present in each model. Additionally, both lines are engineered to express a mutant estrogen receptor binding domain fused to the Zta protein. While this model is useful for studying lytic reactivation, we recognize that it is not a true physiologic trigger for lytic reactivation. To add breadth to our observations, we compared Z-HT to stimulation of the Akata BL cell line with anti-Ig, which leads to reactivation through BCR signaling. While reactivation efficiency was relatively low in the Akata system, we confirmed our overarching findings in all three models.

Given these findings, subsequent examinations of the epigenetic consequences of early EBV reactivation at high resolution should be prioritized, and the possibility of double-edged consequences of oncolytic therapies should be specifically examined in detail. Future single-cell approaches should interrogate the frequency of viable abortive lytic cells [152] and the particular changes in chromatin accessibility as well as other epigenetic features of this phenotype. Similar experimental approaches should be applied to study clinical EBV+ tumor specimens to understand oncogenic correlates of lytic reactivation *in situ*.

## Materials and methods

### Ethics statement

Human cell lines used in this study were not accompanied with HIPAA identifiers or PHI. All experiments were thus categorized as non-human subjects research and approved by a Duke University IRB "Mechanisms of Epstein-Barr virus transformation using adult peripheral blood from the Red Cross" (eIRB #Pro00006262).

### Cell lines, culture, and treatments

P3HR1-ZHT cells (derived from the Type 2 EBV+ [P3 strain] Jijoye eBL line) and B958-ZHT (a marmoset lymphoblastoid cell line transformed with Type 1 EBV [B95-8 strain]) were used in this study. Each cell line was cultured at 37˚C with 5% $CO_2$ in RPMI + 10% FBS (R10) media (Gibco RPMI 1640, ThermoFisher). To induce lytic gene expression, $4x10^5$ cells/mL for a given cell line in log-phase growth were treated with 25 nM, 50 nM, or 100 nM 4-hydroxytamoxifen (4HT) in methanol (4HT, Millipore Sigma). Phosphonoacetic acid (PAA, 1 μM) was included in parallel with lytic induction treatments to inhibit viral DNA synthesis and prevent complete reactivation in separate experimental groups (i.e., abortive lytic replication). Control groups were prepared via treatment with 0.1% DMSO (and DMSO + PAA). All treatments for flow cytometry and RNA Flow-FISH experiments described below were performed in triplicate (technical replicates) in 6-, 12-, or 24-well culture plates.

### Flow cytometry

Flow cytometric cell cycle analysis of unstimulated and 4HT-treated P3HR1-ZHT cells was performed using pulsed BrdU incorporation (20 min) and nuclear staining with 7-AAD in fixed cells (Invitrogen eBioscience BrdU staining kit, cat #8811-6600-42; 7-AAD, cat #00-6993-50) in addition to surface staining for gp350 (mouse anti-gp350 antibody clone 72A1 prepared in house then conjugated to Alexa 647 by Columbia Biosciences). Mitochondrial content versus gp350 expression in 4HT-induced cells was assayed using MitoTracker Green (ThermoFisher, cat #M46750). Flow cytometry was also used to assay surface expression of gp350, CD44, CD133 (PROM1), and CD166 (ALCAM). With the exception of the gp350 antibody, antibodies were purchased from BioLegend (anti-CD44_FITC, cat #397517; anti-CD133_PE, cat #397903; anti-CD166_PE-Cy7, cat #343911). In these experiments, removal of lytic inducing and control treatments at 6, 12, or 24 h via media replacement all yielded similar results. Cell were also stained and gated by viability (ZombieAqua, ThermoFisher, cat #L34965).

### RNA Flow-FISH

RNA Flow FISH analysis of unstimulated and 4HT-induced P3HR1-ZHT cells (24 and 48 h post-treatment) was performed using RNA PrimeFlow reagents (ThermoFisher RNA PrimeFlow Kit Catalog #: 88-18005-210) and validated RNA probes (ThermoFisher. Type 1 probes:

BLLF1_A647. Type 4 probes: BZLF1_A488, BCL2A1_A488. Type 10 probes: BGLF4_A568, CD38_A568, ALDH1A1_A568, SOX2_A568). PrimeFlow sample preparation was completed per ThermoFisher protocol with no adjustments. Briefly, cells were washed, fixed, and permeabilized. Cells were then incubated with target probes for 2 h in a 40˚C water bath. Cells were washed and stored overnight at 4˚C and then incubated with a Pre-amplification buffer for 1.5 h in a 40˚C water bath followed by a 1.5 h incubation in amplification buffer. Cells were then incubated in label probes for 1 h in a 40˚C water bath, washed with FACS buffer and subsequently analyzed on a Cytek Aurora. Spectral flow unmixing was performed with SpectroFlo software and uniformly applied to all samples. Further analysis and gating was completed in FlowJo.

## Single-cell sample and library preparation

P3HR1-ZHT cells were plated at $4 \times 10^5$ cells/ mL in 5 mL R10 then treated with methanol (mock- 0 h) or with 25 nM 4HT (4-hydroxytamoxifen). The cells incubated in 4HT for 72, 48, and 24 hours then all cells were harvested for library preparation at the same time. The viabilities of the 0, 24, 48, and 72 h samples at time of collection were approximately 90%, 80%, 75%, and 75%, respectively. Harvested cells were resuspended at the recommended concentration to collect approximately 10,000 cells per sample during GEM generation. Single-cell transcriptomes from all four samples were captured and reverse transcribed into cDNA libraries using the 10x Genomics Chromium Next GEM Single Cell 3' gene expression kit with v3.1 chemistry and Chromium microfluidic controller according to recommended protocols (10x Genomics, Pleasanton, CA). All cDNA gene expression libraries were pooled for sequencing.

## Sequencing, read alignment, and QC

Pooled single-cell libraries were sequenced across two lanes of an S2 flow cell on a Nova-Seq6000 (Illumina, San Diego, CA) with 50 bp paired-end reads at a target sequencing depth of 50,000 reads per cell. Output base calls (.bcl) were assembled into sample-demultiplexed reads (.fastq) using *cellranger mkfastq* with default settings (10x Genomics, Pleasanton, CA). Reads were mapped to a concatenated reference genome package (hg38 + NC_009334 [type 2 EBV]; prepared via *cellranger mkref*) to generate single-cell expression matrices by running *cellranger count* (10x Genomics, Pleasanton, CA). Cellranger output files (*genes.tsv, barcodes. tsv, matrix.mtx*) were used to create Seurat data objects in R [153–155], which were subsequently pre-processed using QC filters. Cells and features were included if they met the following criteria: feature (gene) expression in a minimum of three cells; mitochondrial genes accounting for $< 25\%$ of all transcripts; a minimum of 200 unique expressed genes; $< 65,000$ total transcripts to exclude non-singlets. The elevated mitochondrial transcript and total transcript cutoffs relative to those used for resting PBMC samples [156] were chosen because of the highly proliferative nature of the P3HR1 cell line, the expectation of apoptosis as one outcome to lytic reactivation, and the implementation of viability enrichment prior to library preparation described above. A total of 26,728 cells across the timecourse passed all QC filters ($n_{untreated} = 10,196$; $n_{24h} = 7,905$; $n_{48h} = 5,841$; $n_{72h} = 3146$).

## Data pre-processing, dropout imputation, analysis, and visualization

A complete list of loaded packages and versions (RStudio *sessionInfo()* output) is provided as a supplementary file (**S1 File**). Single-cell expression data were analyzed and visualized with R (v4.0.5) / RStudio (v2022.07.1+554) using Seurat v4.1.0. Data from each timepoint were analyzed, separated, and merged into a single object to support time-resolved analysis. Raw count data were normalized and scaled prior to feature identification (*NormalizeData* and *ScaleData*

functions). Cell cycle scores and phases were assigned based on annotated gene sets provided in the Seurat package (*CellCycleScoring* function). Expression data were dimensionally reduced using principal component analysis of identified variable features (*RunPCA*), and the first 30 principal components were used for subsequent UMAP dimensional reduction (*FindNeighbors*, *RunUMAP*). Cell clusters were identified at multiple resolutions for phenotype identification and comparative analysis (*FindClusters*).

Biological zero-preserving imputation was applied to correct technical read dropout using adaptive low-rank approximation (ALRA) of the RNA count matrix [157]. Data presented throughout this study was generated from imputed read data. Differential gene expression analysis of the merged timecourse RNA and imputed (ALRA) assays was performed at multiple clustering resolutions. Outputs from this analysis are provided as supplementary tables (S1–S4 Tables). Single-cell gene expression, co-expression, and cluster-averaged expression were visualized with Seurat functions (e.g., *DimPlot*, *FeaturePlot*, *FeatureScatter*, *VlnPlot*, *DotPlot*, *DoHeatmap*). Additional visualization of multi-gene co-expression was generated with the UpSetR package [158].

### Pseudotime analysis

Pseudotime trajectories were calculated for day 0 and merged timecourse datasets using Monocle3 [111,159]. Briefly, Seurat objects were adapted as cell dataset objects and used to learn and order cells along pseudotime graphs anchored at manually determined root cells. Calculated pseudotime values were added as a feature to original Seurat objects and used for subsequent gene expression analyses. Pseudotime-gene correlation was plotted and fit via smoothing splines to visualize expression dynamics across clusters (cell phenotypes).

### Gene ontology and gene regulatory network analyses

Low and high-resolution cluster gene ontology (GO) enrichment analysis for biological processes was performed using the *enrichGO* function in clusterProfiler [90]. Statistically significant enrichment results were visualized using the *barplot*, *pairwise_termism*, and *emapplot* functions. Cluster-level gene regulatory network (GRN) inference of transcription factor activities was conducted using CollecTRI in the R package decoupleR [160,161].

### Statistical analyses

Raw and adjusted p values (Bonferroni correction) were calculated and provided for all identified differentially expressed genes from scRNA-seq data (**S1–S4 Tables**). For conventional flow cytometry and RNA flow-FISH experiments, statistically significant differences between treatment groups were determined via two-tailed Welch's t test (n = 3 replicates per condition).

### Supporting information

**S1 Fig. Flow cytometry replicates for gp350 expression in P3HR1-ZHT cells.** (A) Lymphocyte, singlet, live-cell, and gp350[+] gating for unstimulated cells. (B) The same gating strategy as above applied for 4HT-treated cells. (C) The same gating strategy as above applied for cells co-treated with 4HT and PAA.
(TIF)

**S2 Fig. RNA Flow-FISH replicates for IE, early, and late lytic gene expression in P3HR1-ZHT cells.** (A) Co-expression of *BZLF1* with *BGLF4* or *BLLF1* in DMSO control treatment and 4HT-induced reactivation. (B) Co-expression of *BZLF1*, *BGLF4*, and *BLLF1* (red

cells) in DMSO control treatment and 4HT-induced reactivation.
(TIF)

**S3 Fig. Dot plot of cluster-resolved EBV expression annotated by latent and lytic genes.**
(TIF)

**S4 Fig. Cell cycle and mitochondrial features of P3HR1-ZHT cells.** (A) Cell cycle phase
annotations in P3HR1-ZHT scRNA-seq data. (B) Flow cytometry cell cycle analysis in unstimulated and 4HT-treated P3HR1-ZHT cells with gp350+ cells highlighted. (C) MitoTracker
staining by gp350 status in 4HT-treated P3HR1-ZHT cells.
(TIF)

**S5 Fig. Transcription factor activity prediction in abortive P3HR1-ZHT cells.**
(TIF)

**S6 Fig. Extended dot plot of EBV transcripts expressed across all clusters.**
(TIF)

**S7 Fig. RNA Flow-FISH replicates for *CD38*, *BCL2A1*, and *BLLF1* expression in
P3HR1-ZHT cells.** (A) Technical controls, 24 h, and 48 h responses to DMSO, 4HT, and 4HT
+PAA for *BCL2A1* versus *BLLF1* expression. (B) Technical controls, 24 h, and 48 h responses
to DMSO, 4HT, and 4HT+PAA for *CD38* versus *BLLF1* expression. (C) Technical controls, 24
h, and 48 h responses to DMSO, 4HT, and 4HT+PAA for *BCL2A1* versus *CD38* expression.
(TIF)

**S8 Fig. Quantification and statistical analysis of gp350+ cell frequencies in P3HR1-ZHT
dependent on 4HT-induced reactivation, PAA inhibition of viral DNA synthesis, and NF-
κB pathway inhibition.** Statistical comparisons between groups (n = 3 replicates per treatment
condition) were evaluated via Welch's two-tailed t tests (***p<0.001)
(TIF)

**S9 Fig RNA. Flow-FISH replicates for *ALDH1A1* and *SOX2* expression in *BZLF1*+
P3HR1-ZHT cells with and without 4HT treatment.**
(TIF)

**S10 Fig. Prediction of transcription factor activity associated with reprogrammed pluripotency in lytic P3HR1-ZHT cells.** (A) SOX2 scRNA-seq expression and gene regulatory network activity. (B) Hierarchical clustering of predicted TF activities by P3HR1-ZHT subcluster.
(C) Flow plots of untreated and treated P3HR1-ZHT cells 24 h post stimulation. Flow plots
show that a small percentage of induced cells express SOX2 and that most of these cells are
lytic (BMRF1+). (D) Bar graph depicting the percentage of cells that expressed SOX2 between
treatment groups, in biological triplicate.
(TIF)

**S11 Fig. Flow cytometry replicates for CD44, CD133 (PROM1), and CD166 (ALCAM)
expression in P3HR1-ZHT cells.** (A) Controls, gating, and stemness biomarker expression by
gp350 status in unstimulated cells. (B) Controls, gating, and stemness biomarker expression by
gp350 status in 4HT-treated cells. (C) Controls, gating, and stemness biomarker expression by
gp350 status in cells co-treated with 4HT and PAA.
(TIF)

**S12 Fig. Flow cytometry replicates for gp350 expression in B958-ZHT cells.** (A) Controls,
gating, and gp350 expression in unstimulated cells. (B) Controls, gating, and gp350 expression
in 4HT-treated cells. (C) Controls, gating, and gp350 expression in cells co-treated with 4HT

and PAA.
(TIF)

**S13 Fig. Flow cytometry replicates for CD44, CD133 (PROM1), and CD166 (ALCAM) expression inB958-ZHT cells.** (A) Controls, gating, and stemness biomarker expression by gp350 status in unstimulated cells. (B) Controls, gating, and stemness biomarker expression by gp350 status in 4HT-treated cells. (C) Controls, gating, and stemness biomarker expression by gp350 status in cells co-treated with 4HT and PAA.
(TIF)

**S14 Fig. Independent scRNA-seq replicate validation of key heterogeneous responses in P3HR1-ZHT cells.** (A) Overview of P3HR1-ZHT replicate experiment treatments (methanol control and 4HT) and identified clusters. (B) UMAP visualization of global QC metrics (top row), differential abortive and lytic responses correlated with *STAT3* and *MYC* levels (2nd and 3rd rows), and upregulated pluripotency signature in lytic cell subsets (4th and 5th rows).
(TIF)

**S1 Table. Differentially expressed genes by cluster.** List of differentially expressed genes separated by each unique cluster. Each sheet is a separate cluster and includes gene name, average $\log_2$ fold change, percentages of cells expressing the gene in the cluster versus all other clusters, and adjusted p value.
(XLSX)

**S2 Table. Differentially expressed genes by cluster imputed with ALRA.** List of differentially expressed genes separated by each unique cluster and data was imputed with ALRA in Seurat. Each sheet is a separate cluster and includes gene name, average $\log_2$ fold change, percentages of cells expressing the gene in the cluster versus all other clusters, and adjusted p value.
(XLSX)

**S3 Table. Gene ontology by cluster.** List of GO terms separated by each unique cluster. Each sheet is a separate cluster and includes GO ID, description, adjusted p value, counts, and gene IDs.
(XLSX)

**S4 Table. Gene ontology by cluster imputed with ALRA.** List of GO terms separated by each unique cluster and data was imputed with ALRA in Seurat. Each sheet is a separate cluster and includes GO ID, description, adjusted p value, and gene IDs.
(XLSX)

**S1 File. List of R packages and versions used in the analysis of scRNAseq data.**
(R)

## Acknowledgments

We would like to acknowledge the Heaton Lab at Duke for their assistance with preparing the 10X single cell libraries. We also thank the staff of the Duke Center for Genomic and Computational Biology (GCB) for sequencing support.

## Author Contributions

**Conceptualization:** Elliott D. SoRelle, Heather Christofk, Micah A. Luftig.

**Data curation:** Elliott D. SoRelle, Lauren E. Haynes.

**Formal analysis:** Elliott D. SoRelle, Lauren E. Haynes, Katherine A. Willard, Beth Chang, James Ch'ng.

**Funding acquisition:** Heather Christofk, Micah A. Luftig.

**Investigation:** Elliott D. SoRelle, Lauren E. Haynes, Katherine A. Willard, Beth Chang, James Ch'ng.

**Methodology:** Elliott D. SoRelle.

**Resources:** Heather Christofk.

**Supervision:** Heather Christofk, Micah A. Luftig.

**Visualization:** Elliott D. SoRelle, Lauren E. Haynes.

**Writing – original draft:** Elliott D. SoRelle.

**Writing – review & editing:** Elliott D. SoRelle, Lauren E. Haynes, Katherine A. Willard, James Ch'ng, Heather Christofk, Micah A. Luftig.

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
