## [Decision Letter · Decision Letter 0]

11 Jul 2024

Dear Dr. Luftig,

Thank you very much for submitting your manuscript "Epstein-Barr virus reactivation induces divergent abortive, reprogrammed, and host shutoff states by lytic progression" for consideration at PLOS Pathogens. As with all papers reviewed by the journal, your manuscript was reviewed by members of the editorial board and by several independent reviewers. In light of the reviews (below this email), we would like to invite the resubmission of a significantly-revised version that takes into account the reviewers' comments.

Three referees have reviewed the manuscript, each of whom appreciated strengths of the study, but each of whom have specific, focused points that will be important to address in a revised submission. In particular, reviewer #3 suggests additional supporting evidence from an additional model system (separate from the conditional P3HR-1 model).

We cannot make any decision about publication until we have seen the revised manuscript and your response to the reviewers' comments. Your revised manuscript is also likely to be sent to reviewers for further evaluation.

Sincerely,

Benjamin E Gewurz, M.D., Ph.D.

Academic Editor

PLOS Pathogens

Robert Kalejta

Section Editor

PLOS Pathogens

Michael Malim

Editor-in-Chief

PLOS Pathogens

orcid.org/0000-0002-7699-2064

Three referees have reviewed the manuscript, each of whom appreciated strengths of the study, but each of whom have specific, focused points that will be important to address in a revised submission. In particular, reviewer #3 suggests additional supporting evidence from an additional model system (separate from the conditional P3HR-1 model).

Reviewer's Responses to Questions

**Part I - Summary**

Reviewer #1: This manuscript from the Luftig lab uses single cell analysis to profile changes in B cells that occur upon reactivation of EBV from latent to lytic infection. This is an important question since EBV-positive B cells do not efficiently reactivate, but the reasons for this have not been clear. Using single cell analysis, they have been able to determine correlations between full reactivation and expression of various cellular proteins, suggesting which cellular proteins suppress reactivation. The results support previous studies that Myc suppress reactivation and that full EBV reactivation results in cell cycle arrest in early S. It also provides additional information, including upregulation of NF-kB and IRF3 occurring in abortive lytic cells and correlations of EBV protein expression with host shut-off. Finally, this study identifies stages of reactivation based on expression of EBV genes, showing that some cells enter the lytic cycle partly but not fully, referred to as abortive lytic infection. This is an interesting concept since previous dogma suggested that expression of EBV immediate early genes would be sufficient to activate full lytic infection. I only have minor comments for improvement.

Reviewer #2: This study attempts to address a very interesting mystery in gammaherpesvirus biology, the impact of the host cell environment on the fate of productive viral replication. The authors performed single cell sequencing of two EBV+ B cell lymphoma cell lines, both during latency and following lytic induction. Cells at different stages of lytic replication were identified and pseudotime analysis was performed to interpret the dynamics of host cell gene expression at and between different stages of lytic progression. The authors identify subpopulations of fully lytic cells with a cancer stem cell gene expression program, as well as cells that are a. refractory to lytic entry and b. abortive following lytic entry. Each subpopulation is studied to better understand the underlying host gene expression programs that influence the fate of induced cells. In addition to the important conclusions drawn from this analysis, the data has the potential to serve as a valuable resource for researchers in the field.

The key conclusions of this work, however, hinge on accurate identification of refractory and abortive cells. Complementary approaches to validate subpopulations will provide more convincing evidence of their existence and lead to a more confident interpretation of their underlying characteristics.

Reviewer #3: The manuscript uses scRNAseq methods to analyze the viral and host gene and protein expression patterns upon reactivation of EBV latency. The study focuses primarly on one highly engineered model P3HR1-ZHT where the BZLF1 immediate early transcriptional activator is constitutively expressed and under the regulation of 4-hydroxytamoxifen.

They identify 5 different response clusters. They find that abortive lytic cells have high NF-kB activity, lytic cells with stem-like pluripotency features, and genes that escape the transcriptional host-shut off during lytic reactivation. Finally, they provide corroborating data using a second model utilizing the B958-ZHT.

**Part II – Major Issues: Key Experiments Required for Acceptance**

Reviewer #1: (No Response)

Reviewer #2: - Cells are called refractory if they don’t express lytic genes following induction with 4HT. The authors detect high MYC expression in cells considered refractory, confirming a previous study (Guo et. al., 2020) that MYC expression prevents lytic reactivation. At the same time, MYC expression is reduced by host shut off after lytic commitment, which would also explain finding relatively low MYC levels in reactivating cells. Is the stochasticity of the system an explanation for the population that remain latent? Or is there a population of cells that are inherently refractive to reactivation, and cannot efficiently be induced?

- Cluster C was labeled as “abortive” based on early lytic gene expression. It is unclear if these cells were in a final abortive state or if at the time of collection, they were in an early reactivation stage and would have continued their progression to viral replication. An experiment that follows the trajectory of cells deemed abortive would be helpful to validate the identification of this population.

- The authors find that a number of cancer related genes are induced in a cluster of cells with late lytic gene expression (Figure 4) and suggest the potential for fully lytic cells to be reprogrammed. They confirm expression using RNA Flow-FISH. However, during reactivation, others have reported substantial cellular transcription alterations, many of which are induced unproductive, non-coding transcripts which overlap annotated genes. It would be important to 1. demonstrate protein expression of SOX2 and/or ALDHA1 in late lytic subpopulations, and 2. show evidence that these subpopulations survive lytic replication.

Reviewer #3: Overall

This is an impressive and impactful study that identifies new gene pathways, and confirms previously identified pathways implicated in the control of EBV lytic-latent balance. The application of single cell RNA-seq and Flow FISH are technically strong and appropriate for this type of study. The major limitation and concern is that the data relies primarily, if not exclusively, on one highly engineered model system using P3HR1 BL cell and viral strain, and the BZLF1 conditional ER fusion gene. How well does this system capture critical regulation of BZLF1 and other aspects of physiological cell response to various reactivation stimuli? This limitation should be addressed more directly and extensively in the Discussion section. The authors should consider providing additional experimental evidence that some of these major findings are also true in natural reactivation systems, such as anti-Ig stimulated Akata or other cell systems, particularly non-engineered LCLs or other EBV lymphoid tumor cell lines.

Further, the B958-ZHT cell line is stated to be a marmorset line so it is difficult to understand how the genes are mapped to the human genome effectively.

Specific Comments

Fig 6. Relating to the analysis of the B958-ZHT model as a corroborating system for EBV reactivation. First, this is a marmoset cell line, so that is a question of concern. Second, the activation of BZLF1 remains through the 5-HT system, which may be very different than natural physiological signals. Perhaps related to this, is that the UMAP for the B958-ZHT looks very different than that for the P3HR1-ZHT. While many of the gene correlations are conserved, the authors should also highlight the aspects that are different between these two cell and viral types, and why.

**Part III – Minor Issues: Editorial and Data Presentation Modifications**

Reviewer #1: 1. The authors conclude that clusters C and D are incomplete reactivation states, as compared to cluster E, which is complete lytic infection, and refer to this as abortive lytic infection. However, I am unclear as to exactly which EBV genes are turned on in these clusters, and whether clusters C and D only express some immediate early and early genes without late gene, as suggested by the name abortive lytic. It appears that there is some BLLF1 (gp350) detected in C and D, indicating some late protein expression. Figure 2D is a very nice presentation of some of the changes occurring in the different clusters but only shows 5 EBV genes, whereas I assume most EBV genes would have been detected in cluster E. To better understand the different stages/subtypes of EBV lytic infection, it would be informative to include a representation similar to Fig 2D that shows all the EBV transcripts detected.

2. The authors discuss their single cell profiles in relation to host shutoff and genes that escape host shutoff. However, it is not clear to me what they are using as a measure of host shutoff itself. Is this total cellular mRNA levels? This needs more explanation. Also they state that BGLF5 expression correlates with host shutoff, in keeping with a previously reported role of this protein in shutoff. However there is evidence that other EBV proteins might also play a role in host shutoff. Was BGLF5 the only EBV transcript that correlated with host shutoff in their data? If not a more complete analysis of the relationship of EBV proteins to host shutoff should be presented.

Reviewer #2: - The authors use PAA to model an abortive lytic cellular environment. Although PAA treatment blocks lytic progression, it is unlikely to mimic the underlying cellular environment that promotes abortive lytic replication.

- It would have helped to have had both the raw sequencing data and a gene expression matrix from the single cell sequencing experiments

Reviewer #3: Lines 103-119. References may need to be re-aligned with statements.

PLOS authors have the option to publish the peer review history of their article (what does this mean?). If published, this will include your full peer review and any attached files.

Reviewer #1: **Yes: **Lori Frappier

Reviewer #2: No

Reviewer #3: No
---

## [Decision Letter · Decision Letter 1]

2 Oct 2024

Dear Dr. Luftig,

We are pleased to inform you that your manuscript 'Epstein-Barr virus reactivation induces divergent abortive, reprogrammed, and host shutoff states by lytic progression' has been provisionally accepted for publication in PLOS Pathogens.

Best regards,

Benjamin E Gewurz, M.D., Ph.D.

Academic Editor

PLOS Pathogens

Robert Kalejta

Section Editor

PLOS Pathogens

Michael Malim

Editor-in-Chief

PLOS Pathogens

orcid.org/0000-0002-7699-2064

Reviewer Comments (if any, and for reference):

Reviewer's Responses to Questions

**Part I - Summary**

Reviewer #1: The authors have adequately responded to my previous comments so I think this manuscript is now suitable for publication in Plos Pathogens.

Reviewer #2: (No Response)

**Part II – Major Issues: Key Experiments Required for Acceptance**

Reviewer #1: (No Response)

Reviewer #2: (No Response)

**Part III – Minor Issues: Editorial and Data Presentation Modifications**

Reviewer #1: (No Response)

Reviewer #2: (No Response)

PLOS authors have the option to publish the peer review history of their article (what does this mean?). If published, this will include your full peer review and any attached files.

Reviewer #1: **Yes: **Lori Frappier

Reviewer #2: No

---

## [Editor Report · Acceptance letter]

18 Oct 2024

Dear Dr. Luftig,

We are delighted to inform you that your manuscript, "Epstein-Barr virus reactivation induces divergent abortive, reprogrammed, and host shutoff states by lytic progression," has been formally accepted for publication in PLOS Pathogens.

Best regards,

Michael Malim

Editor-in-Chief

PLOS Pathogens

orcid.org/0000-0002-7699-2064